# Learning Shared Safety Constraints from Multi-task Demonstrations

**Konwoo Kim**[*]
Carnegie Mellon University

**Gokul Swamy**[*]
Carnegie Mellon University

**Zuxin Liu**
Carnegie Mellon University

**Ding Zhao**
Carnegie Mellon University

**Sanjiban Choudhury**
Cornell University

**Zhiwei Steven Wu**
Carnegie Mellon University

## Abstract

Regardless of the particular task we want them to perform in an environment, there are often shared *safety constraints* we want our agents to respect. For example, regardless of whether it is making a sandwich or clearing the table, a kitchen robot should not break a plate. Manually specifying such a constraint can be both time-consuming and error-prone. We show how to learn constraints from expert demonstrations of safe task completion by extending inverse reinforcement learning (IRL) techniques to the space of constraints. Intuitively, we learn constraints that forbid highly rewarding behavior that the expert could have taken but chose not to. Unfortunately, the constraint learning problem is rather ill-posed and typically leads to overly conservative constraints that forbid all behavior that the expert did not take. We counter this by leveraging diverse demonstrations that naturally occur in multi-task settings to learn a tighter set of constraints. We validate our method with simulation experiments on high-dimensional continuous control tasks.

## 1 Introduction

If a friend was in your kitchen and you told them to "make toast" or "clean the dishes," you would probably be rather surprised if they broke some of your plates during this process. The underlying *safety constraint* that forbids these kinds of behavior is both *a)* implicit and *b)* agnostic to the particular task they were asked to perform. Now, let's bring a household robot into the equation, operating within your kitchen. How can we ensure that it adheres to these implicit safety constraints, regardless of its assigned tasks?

One approach might be to write down specific constraints (e.g. joint torque limits) and pass them to the decision-making system of the robot. Unfortunately, more complex constraints like the ones we consider above are both difficult to formalize mathematically and easy for an end-user to forget to specify (as they would be inherently understood by a human helper). This problem is paralleled in the field of reinforcement learning (RL), where defining reward functions that lead to desirable behaviors for the learning agent is a recurring challenge [Hadfield-Menell et al., 2017]. For example, it is rather challenging to handcraft the exact function one should be optimized to be a good driver. The standard solution to this sort of "reward design" problem is to instead demonstrate the desired behavior of the agent and then extract a reward function that would incentivize such behavior. Such *inverse reinforcement learning* (IRL) techniques have found application in fields as diverse as robotics [Silver et al., 2010, Ratliff et al., 2009, Kolter et al., 2008, Ng et al., 2006, Zucker et al., 2011], computer vision [Kitani et al., 2012], and human-computer interaction [Ziebart et al., 2008b, 2012]. Given the success of IRL techniques and the similarity between reward and constraint design, we propose

---

[*]Equal contribution. Correspondence to `gswamy@cmu.edu`.

37th Conference on Neural Information Processing Systems (NeurIPS 2023).

extending IRL techniques to the space of constraints. We term such techniques *inverse constraint learning*, or ICL for short.

More formally, we consider a setting in which we have access to demonstrations of an expert policy for a task, along with knowledge about the task's reward. This allows us to look at the difference between the expert and reward-optimal policies for a task. Our first key insight is that ***the actions taken by the reward-optimal but not the expert policy are likely to be forbidden, allowing us to extract a constraint.***

Unfortunately, the ICL problem is still rather ill-posed. Indeed, prior work in ICL will often learn overly conservative constraints that forbid all behavior the expert did not take [Scobee and Sastry, 2019, Vazquez-Chanlatte et al., 2018, McPherson et al., 2021]. However, for tasks in a shared environment with different rewards, there are often safety constraints that should be satisfied regardless of the task (e.g. a plate shouldn't be broken regardless of whether you're serving food on it or cleaning up after a meal). Our second crucial insight is that ***we can leverage multi-task data to provide more comprehensive demonstration coverage over the state space, helping our method avoid degenerate solutions.***

More explicitly, the contributions of our work are three-fold.

**1. We formalize the inverse constraint learning problem.** We frame ICL as a zero-sum game between a policy player and a constraint player. The policy player attempts to maximize reward while satisfying a potential constraint, while the constraint player picks constraints that maximally penalize the learner relative to the expert. Intuitively, such a procedure recovers constraints that forbid high-reward behavior the expert did not take.

**2. We develop a multi-task extension of inverse constraint learning.** We derive a zero-sum game between a set of policy players, each attempting to maximize a task-specific reward, and a constraint player that chooses a constraint that all policy players must satisfy. Because the constraint player looks at aggregate learner and expert data, it is less likely to select a degenerate solution.

**3. We demonstrate the efficacy of our approach on various continuous control tasks.** We show that with restricted function classes, we are able to recover ground-truth constraints on certain tasks. Even when using less interpretable function classes like deep networks, we can still ensure a match with expert safety and task performance. In the multi-task setting, we are able to identify constraints that a single-task learner would struggle to learn.

We begin with a discussion of related work.

## 2  Related Work

Our work exists at the confluence of various research thrusts. We discuss each independently.

**Inverse Reinforcement Learning.** IRL [Ziebart et al., 2008a,b, 2012, Ho and Ermon, 2016] can be framed as a two-player zero-sum game between a policy player and a reward player [Swamy et al., 2021]. In most formulations of IRL, a potential reward function is chosen in an outer loop, and the policy player optimizes it via RL in an inner loop. Similar to IRL, the constraint in our formulation of ICL is chosen adversarially in an outer loop. However, in contrast to IRL, the inner loop of ICL is *constrained* reinforcement learning: the policy player tries to find the optimal policy that respects the constraint chosen in the outer loop.

**Constrained Reinforcement Learning.** Our approach involves repeated calls to a constrained reinforcement learning (CRL) oracle [García and Fernández, 2015, Gu et al., 2022]. CRL aims to find a reward-maximizing policy over a constrained set, often formulated as a constrained policy optimization problem [Altman, 1999, Xu et al., 2022]. Solving this problem via Frank-Wolfe methods is often unstable [Ray et al., 2019, Liang et al., 2018]. Various methods have been proposed to mitigate this instability, including variational techniques [Liu et al., 2022], imposing trust-region regularization [Achiam et al., 2017, Yang et al., 2020, Kim and Oh, 2022], optimistic game-solving algorithms [Moskovitz et al., 2023], and PID controller-based methods [Stooke et al., 2020]. In our practical implementations, we use PID-based methods for their relative simplicity.

**Multi-task Inverse Reinforcement Learning.** Prior work in IRL has considered incorporating multi-task data [Xu et al., 2019, Yu et al., 2019, Gleave and Habryka, 2018]. We instead consider a

setting in which we know task-specific rewards and are attempting to recover a shared component of the demonstrator's objective. Amin et al. [2017] consider a similar setting but require the agent to be able to actively choose tasks or interactively query the expert, while our approach requires neither.

**Inverse Constraint Learning.** We are far from the first to consider the ICL problem. Scobee and Sastry [2019], McPherson et al. [2021] extend the MaxEnt IRL algorithm of Ziebart et al. [2008a] to the ICL setting. We instead build upon the moment-matching framework of Swamy et al. [2021], allowing our theory to handle general reward functions instead of the linear reward functions MaxEnt IRL assumes. We are also able to provide performance and constraint satisfaction guarantees on the learned policy, unlike the aforementioned work. Furthermore, we consider the multi-task setting, addressing a key shortcoming of the prior work.

Perhaps the most similar paper to ours is the excellent work of Chou et al. [2020], who also consider the multi-task ICL setting but propose a solution that requires several special solvers that depend on knowledge of the parametric family that a constraint falls into. In contrast, we provide a general algorithmic template that allows one to apply whatever flexible function approximators (e.g. deep networks) and reinforcement learning algorithms (e.g. PPO) they desire. Chou et al.'s method also requires sampling *uniformly* over the set of trajectories that achieve a higher reward than the expert, a task which is rather challenging to do on high-dimensional problems. In contrast, our method only requires the ability to solve a standard RL problem. Theoretically, Chou et al. [2020] focus on constraint recovery, which we argue below is a goal that requires strong assumptions and is therefore a red herring on realistic problems. This focus also prevents their theory from handling suboptimal experts. In contrast, we are able to provide rigorous guarantees on learned policy performance and safety, even when the expert is suboptimal. We include results in Appendix C that show that our approach is far more performant.

In concurrent work, Lindner et al. [2023] propose an elegant solution approach to ICL: rather than learning a constraint function, assume that *any* unseen behavior is unsafe and enforce constraints on the learner to play a convex combination of the demonstrated safe trajectories. The key benefit of this approach is that it doesn't require knowing the reward function the expert was optimizing. However, by forcing the learner to simply replay previous expert behavior, the learner cannot meaningfully generalize, and might therefore be extremely suboptimal on any new task. In contrast, we use the side information of a reasonable set of constraints to provide rigorous policy performance guarantees.[2]

We now turn our attention to formalizing inverse constraint learning.

## 3 Formalizing Inverse Constraint Learning

We build up to our full method in several steps. We first describe the foundational algorithmic structures we build upon (inverse reinforcement learning and constrained reinforcement learning). We then describe the single-task formulation before generalizing it to the multi-task setup.

We consider a finite-horizon Markov Decision Process (MDP) [Puterman, 2014] parameterized by $\langle \mathcal{S}, \mathcal{A}, \mathcal{T}, r, T \rangle$ where $\mathcal{S}$, $\mathcal{A}$ are the state and action spaces, $\mathcal{T} : \mathcal{S} \times \mathcal{A} \to \Delta(\mathcal{S})$ is the transition operator, $r : \mathcal{S} \times \mathcal{A} \to [-1, 1]$ is the reward function, and $T$ is the horizon.

### 3.1 Prior Work: Inverse RL as Game Solving

In the inverse RL setup, we are given access trajectories generated by an expert policy $\pi^E : \mathcal{S} \to \Delta(\mathcal{A})$, but do not know the reward function of the MDP. Our goal is to nevertheless learn a policy that performs as well as the expert's, no matter the true reward function.

We solve the IRL problem via equilibrium computation between a policy player and an adversary that tries to pick out differences between expert and learner policies under potential reward functions Swamy et al. [2021]. More formally, we optimize over polices $\pi : \mathcal{S} \to \Delta(\mathcal{A}) \in \Pi$ and reward functions $f : \mathcal{S} \times \mathcal{A} \to [-1, 1] \in \mathcal{F}_r$. For simplicity, we assume that our strategy spaces ($\Pi$ and $\mathcal{F}_r$) are convex and compact and that $r \in \mathcal{F}_r, \pi_E \in \Pi$. We solve (i.e. compute an approximate Nash

---

[2]We also note that, because we scale the learned constraint differently for each task, Lindner et al. [2023]'s impossibility result (Prop. 2) does not apply to our method, thereby elucidating why a naive application of inverse RL on the aggregate data isn't sufficient for the problem we consider.

---

**Algorithm 1** `CRL` (Constrained Reinforcement Learning)

---

**Input:** Reward $r$, constraint $c$, learning rates $\eta_{1:N}$, tolerance $\delta$
**Output:** Trained policy $\pi$
Initialize $\lambda_1 = 0$
**for** $i$ in $1 \ldots N$ **do**
$\quad \pi_i \leftarrow \mathtt{RL}(r = r - \lambda_i c)$
$\quad \lambda_i \leftarrow [\lambda_i + \eta_i(J(\pi_i, c) - \delta)]^+$
**end for**
**Return** $\mathrm{Unif}(\pi_{1:N})$.

---

**Algorithm 2** `ICL` (Inverse Constraint Learning)

---

**Input:** Reward $r$, constraint class $\mathcal{F}_c$, trajectories from $\pi_E$
**Output:** Learned constraint $c$
Initialize $c_1 \in \mathcal{F}_c$
**for** $i$ in $1 \ldots N$ **do**
$\quad \pi_i, \lambda_i \leftarrow \mathtt{CRL}(r, c_i, \delta = J(\pi_E, c_i))$
$\quad$ `// use any no-regret algo.  to pick c, e.g.  FTRL:`
$\quad c_{i+1} \leftarrow \mathrm{argmax}_{c \in \mathcal{F}_c} \frac{1}{T} \sum_j^i (J(\pi_j, c) - J(\pi_E, c)) - R(c).$
**end for**
**Return** best of $c_{1:N}$ on validation data.

---

equilibrium) of the two-player zero-sum game

$$\min_{\pi \in \Pi} \max_{f \in \mathcal{F}_r} J(\pi, f) - J(\pi_E, f), \tag{1}$$

where $J(\pi, f) = \mathbb{E}_{\xi \sim \pi}[\sum_{t=0}^T f(s_t, a_t)]$ denotes the value of policy $\pi$ under reward function $f$.

## 3.2 Prior Work: Constrained Reinforcement Learning as Game Solving

In CRL, we are given access to both the reward function and a constraint $c : \mathcal{S} \times \mathcal{A} \to [-1, 1]$. Our goal is to learn the highest reward policy that, over the horizon, has a low expected value under the constraint. More formally, we seek a solution to the optimization problem:

$$\min_{\pi \in \Pi} -J(\pi, r) \text{ s.t. } J(\pi, c) \leq \delta, \tag{2}$$

where $\delta$ is some error tolerance. We can also formulate CRL as a game via forming the Lagrangian of the above optimization problem [Altman, 1999]:

$$\min_{\pi \in \Pi} \max_{\lambda > 0} -J(\pi, r) + \lambda(J(\pi, c) - \delta). \tag{3}$$

Intuitively, the adversary updates the weight of the constraint term in the policy player's reward function based on how in violation the learner is.

## 3.3 Single-Task Inverse Constraint Learning

We are finally ready to formalize ICL. In ICL, we are given access to the reward function, trajectories from the solution to a CRL problem, and a class of potential constraints $\mathcal{F}_c$ in which we assume the ground-truth constraint $c^*$ lies. We assume that $\mathcal{F}_c$ is convex and compact.

In the IRL setup, without strong assumptions on the dynamics of the underlying MDP and expert, it is impossible to guarantee recovery of the ground-truth reward. Often, the only reward function that actually makes the expert optimal is zero everywhere [Abbeel and Ng, 2004]. Instead, we attempt to find the reward function that maximally distinguishes the expert from an arbitrary other policy in our policy class via game-solving [Ziebart et al., 2008a, Ho and Ermon, 2016, Swamy et al., 2021]. Similarly, for ICL, exact constraint recovery can be challenging. For example, if two constraints differ only on states the expert never visits, it is not clear how to break ties. We instead try to find a constraint that best separates the safe (but not necessarily optimal) $\pi_E$ from policies that achieve higher rewards.

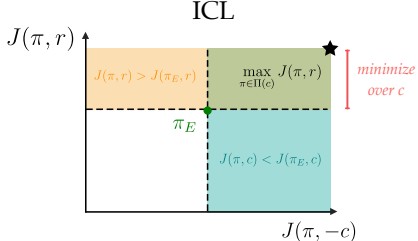

Figure 1: A visual depiction of the optimization problem we're trying to solve in ICL. We attempt to pick a constraint that minimizes the value difference over the expert policy a safe policy could have. The star corresponds to the output of CRL.

More formally, we seek to solve the following constrained optimization problem.

$$\min_{\pi \in \Pi} J(\pi_E, r) - J(\pi, r) \tag{4}$$

$$\text{s.t.} \max_{c \in \mathcal{F}_c} J(\pi, c) - J(\pi_E, c) \leq 0. \tag{5}$$

Note that in contrast to the *moment-matching* problem we solve in imitation learning [Swamy et al., 2021], we instead want to be *at least* as safe as the expert. This means that rather than having equality constraints, we have inequality constraints. Continuing, we can form the Lagrangian:

$$\min_{\pi \in \Pi} \max_{\lambda > 0} J(\pi_E, r) - J(\pi, r) + \lambda(\max_{c \in \mathcal{F}_c} J(\pi, c) - J(\pi_E, c)) \tag{6}$$

$$= \max_{c \in \mathcal{F}_c} \max_{\lambda > 0} \min_{\pi \in \Pi} J(\pi_E, r - \lambda c) - J(\pi, r - \lambda c). \tag{7}$$

Notice that the form of the ICL game resembles a combination of the IRL and CRL games. We describe the full game-solving procedure in Algorithm 2, where $R(c)$ is an arbitrary strongly convex regularizer [McMahan, 2011]. Effectively, we pick a constraint function in the same way we pick a reward function in IRL but run a CRL inner loop instead of an RL step. Instead of a fixed constraint threshold, we set tolerance $\delta$ to the expert's constraint violation. Define

$$\ell_i(c) = \frac{1}{T}(J(\pi_i, c) - J(\pi_E, c)) \in [-1, 1] \tag{8}$$

as the per-round loss that the constraint player suffers in their online decision problem. The best-in-hindsight comparator constraint is defined as

$$\hat{c} = \operatorname*{argmax}_{c \in \mathcal{F}_c} \sum_i^T \ell_i(c). \tag{9}$$

We can then define the cumulative regret the learner suffers as

$$\text{Reg}(T) = \sum_i^T \ell_i(\hat{c}) - \sum_i^T \ell_i(c_i), \tag{10}$$

and let $\epsilon_i = \ell_i(\hat{c}) - \ell_i(c_i)$. We prove the following theorem via standard machinery.

**Theorem 3.1.** *Let $c_{1:N}$ be the iterates produced by Algorithm 2 and let $\bar{\epsilon} = \frac{1}{N} \sum_i^N \epsilon_i$ denote their time-averaged regret. Then, there exists a $c \in c_{1:N}$ such that $\pi = \mathtt{CRL}(r, c, \delta = J(\pi_E, c))$ satisfies*

$$J(\pi, c^*) - J(\pi_E, c^*) \leq \bar{\epsilon}T \text{ and } J(\pi, r) \geq J(\pi_E, r). \tag{11}$$

In words, by optimizing under the recovered constraint, we can learn a policy that (weakly) Pareto-dominates the expert policy under $c^*$. We conclude by noting that because FTRL (Follow the Regularized Leader, McMahan [2011]) is a no-regret algorithm for linear losses like (8), we have that $\lim_{T \to \infty} \frac{\text{Reg}(T)}{T} = 0$. This means that with enough iterations, the RHS of the above bound on ground-truth constraint violation will go to 0.

---
**Algorithm 3** MT-ICL (Multi-task Inverse Constraint Learning)
---
**Input:** Rewards $r^{1:K}$, constraint class $\mathcal{F}_c$, trajectories from $\pi_E^{1:K}$
**Output:** Learned constraint $c$
Set $\widetilde{\mathcal{F}}_c = \{c \in \mathcal{F}_c | \forall k \in [K], J(\pi_E^k, c) \leq 0\}$
Initialize $c_1 \in \widetilde{\mathcal{F}}_c$
**for** $i$ in $1 \dots N$ **do**
    **for** $k$ in $1 \dots K$ **do**
        $\pi_i^k, \lambda_i^k \leftarrow \text{CRL}(r^k, c_i, \delta = 0)$
    **end for**
    `// use any no-regret algo. to pick c, e.g. FTRL:`
    $c_{i+1} \leftarrow \text{argmax}_{c \in \widetilde{\mathcal{F}}_c} \frac{1}{TK} \sum_j^i \sum_k^K (J(\pi_j^k, c) - J(\pi_E^k, c)) - R(c)$.
**end for**
**Return** best of $c_{1:N}$ on validation data.
---

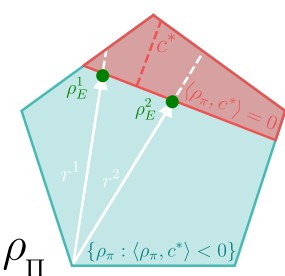

Figure 2: If we have a sufficient diversity of expert policies, none of which are optimal along the reward vector, we can identify the hyperplane that separates the safe policies from the unsafe policies. The constraint (red, dashed) will be orthogonal to this hyperplane. For this example, because $\rho_\pi \in \mathbb{R}^2$, we need two expert policies.

## 3.4 Multi-task Inverse Constraint Learning

One of the potential failure modes of the single-task approach we outline above is that we could learn an overly conservative constraint, leading to poor task performance [Liu et al., 2023]. For example, imagine that we entropy-regularize our policy optimization [Ziebart et al., 2008a, Haarnoja et al., 2018], as is common practice. Assuming a full policy class, the learner puts nonzero probability mass on all reachable states in the MDP. The constraint player is therefore incentivized to forbid all states the expert did not visit [Scobee and Sastry, 2019, McPherson et al., 2021]. Such a constraint would likely generalize poorly when combined with a new reward function ($\tilde{r} \neq r$) as it forbids *all untaken* rather than just *unsafe* behavior.

At heart, the issue with the single-task formulation lies in the potential for insufficient coverage of the state space within expert demonstrations. Therefore, it is natural to explore a multi-task extension to counteract this limitation. Let each task be defined by a unique reward. We assume the dynamics and safety constraints are consistent across tasks. We observe $K$ samples of the form $(r_k, \{\xi \sim \pi_E^k\})$. This data allows us to define the multi-task variant of our previously described ICL game:

$$\max_{c \in \mathcal{F}_c} \min_{\pi^{1:K} \in \Pi} \max_{\lambda^{1:K} > 0} \sum_i^K J(\pi_E^i, r^i - \lambda^i c) - J(\pi^i, r^i - \lambda^i c). \tag{12}$$

We describe how we solve this game in Algorithm 3, where $R(c)$ is an arbitrary strongly convex regularizer [McMahan, 2011]. In short, we alternate between solving $K$ CRL problems and updating the constraint based on the data from all policies.

We now give two conditions under which generalization to new reward functions is possible.

## 3.5 A (Strong) Geometric Condition for Identifiability

Consider for a moment the linear programming (LP) formulation of reinforcement learning. We search over the space of occupancy measures ($\rho_\pi \in \Delta(\mathcal{S} \times \mathcal{A})$) that satisfy the set of Bellman flow constraints [Sutton and Barto, 2018] and try to maximize the inner product with reward vector $r \in \mathbb{R}^{|\mathcal{S}||\mathcal{A}|}$. We can write the CRL optimization problem (assuming $\delta = 0$ for simplicity) as an LP

as well. Using $\rho_\Pi$ to denote the occupancy measures of all $\pi \in \Pi$,

$$\max_{\rho_\pi \in \rho_\Pi} \langle \rho_\pi, r \rangle \text{ s.t. } \langle \rho_\pi, c^* \rangle \leq 0.$$

We observe a (for simplicity, optimal) solution to such a problem for $K$ rewards, begging the question of when that is enough to uniquely identify $c^*$. Recall that to uniquely determine the equation of a hyperplane in $\mathbb{R}^d$, we need $d$ linearly independent points. $c^* \in \mathbb{R}^{|\mathcal{S}||\mathcal{A}|}$, so we need $|\mathcal{S}||\mathcal{A}|$ expert policies. Furthermore, we need each of these points to lie on the constraint line and not on the boundary of the full polytope. Put differently, we need each distinct expert policy to *saturate* the underlying constraint (i.e. $\exists \pi \in \Pi$ s.t. $J(\pi_E^k, r^k) < J(\pi^k, r^k)$). Under these conditions, we can uniquely determine the hyperplane that separates safe from unsafe policies, to which the constraint vector is orthogonal. More formally,

**Lemma 3.2.** *Let $\pi_E^{1:|\mathcal{S}||\mathcal{A}|}$ be distinct optimal expert policies such that a) $\forall i \in [|\mathcal{S}||\mathcal{A}|]$, $\pi_E^i \in relint(\rho_\Pi)$ and b) no $\rho_{\pi_E^i}$ can be generated by a mixture of the other visitation distributions. Then, $c^*$ is the unique (up to scaling) nonzero vector in*

$$Nul\left(\begin{bmatrix} \rho_{\pi_E^1} - \rho_{\pi_E^2} \\ \vdots \\ \rho_{\pi_E^{|\mathcal{S}||\mathcal{A}|-1}} - \rho_{\pi_E^{|\mathcal{S}||\mathcal{A}|}} \end{bmatrix}\right). \tag{13}$$

We visualize this process for the $|\mathcal{S}||\mathcal{A}| = 2$ case in Fig. 2. Assuming we are able to recover $c^*$, we can guarantee that our learners will be able to act safely, regardless of the task they are asked to do. However, the assumptions required to do so are quite strong: we are effectively asking for our expert policies to form a basis for the space of occupancy measures, which means we must see expert data for a large set of diverse tasks. Furthermore, we need the experts to be reward-optimal.

Identifiability (the goal of prior works like Chou et al. [2020], Amin et al. [2017]) is too strong a goal as it requires us to estimate the value of the constraint *everywhere* in the state-action space. If we know the learner will only be incentivized to go to a certain subset of states (as is often true in practice), we can guarantee safety without fully identifying $c^*$. Therefore, we now consider how, by making distributional assumptions on how tasks are generated, we can generalize to novel tasks.

### 3.6 A Statistical Condition for Generalization

Assume that tasks $\tau$ are drawn i.i.d. from some $P(\tau)$. Then, even if we do not see a wide enough diversity of expert policies to guarantee identifiability of the ground-truth constraint function, with enough samples, we can ensure we do well in expectation over tasks. For some constraint $c$, let us define

$$V(c) = \mathbb{E}_{\tau \sim P(\tau)}[J(\pi^\tau, c) - J(\pi_E^\tau, c)], \tag{14}$$

where $\lambda^\tau, \pi^\tau = \text{CRL}(r^\tau, c)$ denote the solutions to the inner optimization problem. We begin by proving the following lemma.

**Lemma 3.3.** *With*

$$K \geq O\left(\log\left(\frac{|\mathcal{F}_c|}{\delta}\right)\frac{(2T)^2}{\epsilon^2}\right) \tag{15}$$

*samples, we have that with probability $\geq 1 - \delta$, we will be able to estimate all $|\mathcal{F}_c|$ population estimates of $V(c)$ within $\epsilon$ absolute error.*

Note that we perform the above analysis for finite classes but one could easily extend it [Sriperumbudur et al., 2009]. The takeaway from the above lemma is that if we observe a sufficient number of tasks, we can guarantee that we can estimate the population loss of all constraints, up to some tolerance.

Consider the learner being faced with a new task they have never seen before at test time. Unlike in the single task case, where it is clear how to set the cost limit passed to CRL, it is not clear how to do so for a novel task. Hence, we make the following assumption.

**Assumption 3.4.** We assume that $\mathbb{E}_\tau[J(\pi_E^\tau, c^*)] \leq 0$, and that $\forall c \in \mathcal{F}_c$, $\exists \pi \in \Pi$ s.t. $J(\pi, c) \leq 0$.

This (weak) assumption allows us to a) use a cost limit of 0 for our CRL step and b) search over a subset of $\mathcal{F}_c$ that the expert is safe under. Under this assumption, we are able to prove the following:

**Theorem 3.5.** *Let $c_{1:N}$ be the iterates produced by Algorithm 3 with $K(\epsilon, \delta)$ chosen as in Lemma 3.3 and let $\bar{\epsilon} = \frac{1}{N} \sum_i^N \epsilon_i$ denote their time-averaged regret. Then, w.p. $\geq 1 - \delta$, there exists a $c \in c_{1:N}$ such that $\pi(r) = \mathcal{CRL}(r, c, \delta = 0)$ satisfies*

$$\mathbb{E}_{\tau \sim P(\tau)}[J(\pi(r^\tau), c^*) - J(\pi_E^\tau, c^*)] \leq \bar{\epsilon}T + 3\epsilon T \text{ and } \mathbb{E}_{\tau \sim P(\tau)}[J(\pi(r^\tau), r^\tau) - J(\pi_E^\tau, r^\tau)] \geq -2\epsilon T.$$

In short, if we observe enough tasks, we are able to learn a constraint that, when optimized under, leads to policies that approximately Pareto-dominate those of the experts on average.

We now turn our attention to the practical implementation of these algorithms.

## 4 Practical Algorithm

We provide practical implementations of constrained reinforcement learning and inverse constraint learning and benchmark their performance on several continuous control tasks. We first describe the environments we test our algorithms on. Then, we provide results showing that our algorithms learn policies that match expert performance and constraint violation. While it is hard to guarantee constraint recovery in theory, we show that we can recover the ground-truth constraint empirically if we search over a restricted enough function class.

### 4.1 Tasks

We focus on the ant environment from the PyBullet [Coumans and Bai, 2016] and MuJoCo [Todorov et al., 2012] benchmarks. The default reward function incentivizes progress along the positive $x$ direction. For our single-task experiments, we consider a velocity and position constraint on top of this reward function.

1. **Velocity Constraint:** $\frac{\|q_{t+1} - q_t\|_2}{dt} \leq 0.75$ where $q_t$ is the ant's position
2. **Position Constraint:** $0.5x_t - y_t \leq 0$ where $x_t, y_t$ are the ant's coordinates

For our multi-task experiments, we build upon the D4RL [Fu et al., 2020] AntMaze benchmark. The default reward function incentivizes the agent to navigate a fixed maze to a random goal position: $\exp(-\|q_{\text{goal}} - q_t\|_2)$. We modify this environment such that the walls of the maze are permeable, but the agent incurs a unit step-wise cost for passing through the maze walls.

Our expert policies are generated by running CRL with the ground-truth constraint. We use the Tianshou [Weng et al., 2022] implementation of PPO [Schulman et al., 2017] as our baseline policy optimizer. Classical Lagrangian methods exactly follow the gradient update shown in Algorithm 1, but they are susceptible to oscillating learning dynamics and constraint-violating behavior during training. The PID Lagrangian method [Stooke et al., 2020] extends the naive gradient update of $\lambda_i$ with a proportional and derivative term to dampen oscillations and prevent cost overshooting. To reduce the amount of interaction required to solve the inner optimization problem, we warm-start our policy in each iteration by behavior cloning against the given expert demonstrations. We used a single NVIDIA 3090 GPU for all experiments. Due to space constraints, we defer all other implementation details to Appendix B.

### 4.2 ICL Results

We begin with results for the single-task problem, before continuing on to the multi-task setup.

### 4.3 Single-Task Continuous Control Results

As argued above, we expect a proper ICL implementation to learn policies that perform as well and are as safe as the expert. However, by restricting the class of constraints we consider, we can also investigate whether recovery of the ground-truth constraint is possible. To this end, we consider a reduced-state version of our algorithm where the learned constraint takes a subset of the agent state as input. For the velocity constraint, the learned constraint is a linear function of the velocity, while for the position and maze constraints, the learned constraint is a linear function of the ant's position.

Using this constraint representation allows us to visualize the learned constraint over the course of training, as shown in Figure 3. We find that our ICL implementation is able to recover the constraint,

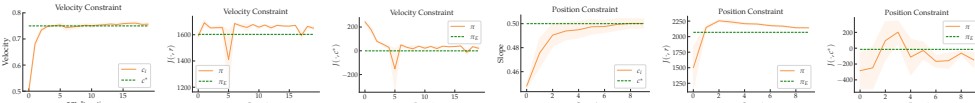

Figure 3: Over the course of training, the learned ICL constraint recovers the ground-truth constraints for the velocity and position tasks. The learned policy matches expert performance and constraint violation. Standard errors are computed across 3 seeds.

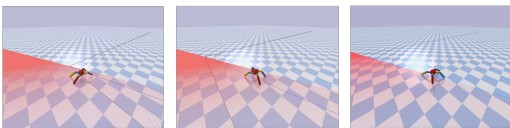

Figure 4: As ICL training progresses, the learned position constraint (red line) converges to the ground-truth constraint (blue line) and the policy learns to escape unsafe regions (red region).

as the learned constraint for both the velocity and position tasks converges to the ground-truth value. Our results further show that over the course of ICL training, the learned policies match and exceed expert performance as their violations of the ground-truth constraint converge towards the expert's. Figure 4 provides a direct depiction of the evolution of the learned constraint and policy. The convergence of the red and blue lines shows that the learned position constraint approaches the ground truth, and the policy's behavior approaches that of the expert in response to this.

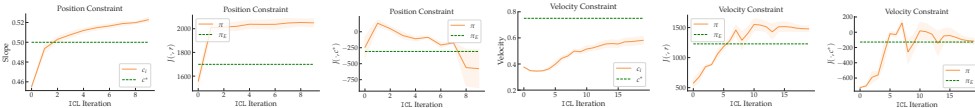

Figure 5: We re-conduct our position velocity constraint experiments using suboptimal experts to generate demonstrations. While, because of the ill-posedness of the problem, we do not exactly recover the ground truth constraint, we are able to use it to learn a policy that is higher performance than the expert while being just as safe. Standard errors are computed across 3 seeds.

To measure the robustness of our method to sub-optimal experts, we repeat the above experiments using demonstrations from an expert with i.i.d. Gaussian noise added to their actions at each timestep. We are still able to learn a safe and performant policy, which matches what our theory predicts.

## 4.4 Multi-Task Continuous Control Results

We next consider an environment where, even with an appropriate constraint class, recovering the ground-truth constraint with a single task isn't feasible due to the ill-posedness of the inverse constraint learning problem. Specifically, we use the umaze AntMaze environment from D4RL [Fu et al., 2020], modified to have a more complex maze structure. As seen in Figure 7, the goal of each task is to navigate through the maze from one of the starting positions (top/bottom left) to one of the grid cells in the rightmost column. We provide expert data for all 10 tasks to the learner.

As we can see in Figure 6, multi-task ICL is, within a single iteration, able to learn policies that match expert performance and constraint violation across all tasks, *all without ever interacting with the ground-truth maze*. Over time, we are able to approximately recover the entire maze structure.

We visually compare several alternative strategies for using the multi-task demonstration data in the bottom row of Figure 7. The 0/1 values in the cells correspond to querying the deep constraint network from the last iteration of ICL on points from each of the grid cells and thresholding at some confidence. We see that a single-task network *(d)* learns spurious walls that would prevent the learner from completing more than half of the tasks. Furthermore, learning 10 separate classifiers and then aggregating their outputs *(e)* / *(f)* also fails to produce reasonable outputs. However, when we use data from all 10 tasks to train our multi-task constraint network *(g)* / *(h)*, we are able to approximately recover the walls of the maze. These results echo our preceding theoretical argument about the importance of multi-task data for learning constraints that generalize to future tasks.

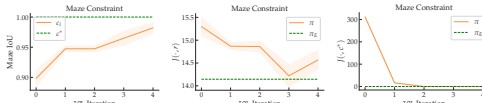

Figure 6: We see that over ICL iterations, we are able to recover the ground-truth walls of the ant-maze, enabling the learner to match expert performance and constraint violations. Results for the second two plots are averaged across all 10 tasks. Standard errors are computed across 3 seeds.

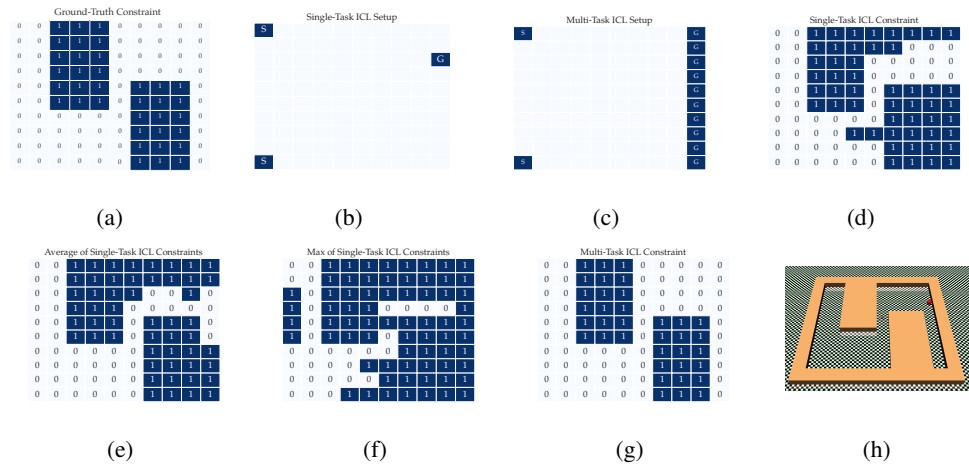

Figure 7: We consider the problem of trying to learn the walls of a custom maze **(a)** based on the AntMaze environment from D4RL [Fu et al., 2020]. We consider both a single-task **(b)** and multi-task **(c)** setup. We see that the single-task data is insufficient to learn an accurate constraint **(d)**. Averaging or taking the max over the constraints learned from the data for each of the ten goals **(e)-(f)** also doesn't work. However, if we use the data from all 10 tasks to learn the constraint **(g)-(h)**, we are able to approximately recover the ground-truth constraint with enough constraint learning iterations.

We release the code we used for all of our experiments at https://github.com/konwook/mticl.

## 5   Discussion

In this work, we derive an algorithm for learning safety constraints from multi-task demonstrations. We show that by replacing the inner loop of inverse reinforcement learning with a constrained policy optimization subroutine, we can learn constraints that guarantee learner safety on a single task. We then give statistical and geometric conditions under which we can guarantee safety on unseen tasks by planning under a learned constraint. We validate our approach on several control tasks.

**Limitations.** In the future, we would be interested in applying our approach to real-world problems (e.g. offroad driving). Algorithmically, the CRL inner loop can be more computationally expensive than an RL loop – we would be interested in speeding up CRL using expert demonstrations, perhaps by adopting the approach of Swamy et al. [2023]. We also ignore all finite-sample issues, which could potentially be addressed via data-augmentation approaches like that of Swamy et al. [2022].

## 6   Acknowledgements

We thank Drew Bagnell for edifying conversations on the relationship between ICL and IRL and Nan Jiang for connecting us to various references in the literature. We also thank an anonymous reviewer for pointing out that our method does not actually require the expert to be the optimal safe policy, a fact we did not fully appreciate beforehand. ZSW is supported in part by the NSF FAI Award #1939606, a Google Faculty Research Award, a J.P. Morgan Faculty Award, a Facebook Research Award, an Okawa Foundation Research Grant, and a Mozilla Research Grant. KK and GS are supported by a GPU award from NVIDIA.

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

# A Proofs

## A.1 Proof of Theorem 3.1

*Proof.* Let $\Pi(c)$ denote the set of policies such that

$$J(\pi, c) - J(\pi_E, c) \leq 0. \tag{16}$$

First, note that $\forall c \in \mathcal{F}_c, \pi_E \in \Pi(c)$. CRL is therefore trying to maximize reward over a set of policies that contains $\pi_E$. Thus, the reward condition is trivially true. We therefore focus on the safety condition. First, we note that by the definition of regret,

$$\frac{1}{N} \sum_i^N [(J(\pi_i, \hat{c}) - J(\pi_E, \hat{c})) - (J(\pi_i, c_i) - J(\pi_E, c_i))] = \frac{T}{N} \sum_i^N \ell_i(\hat{c}) - \ell_i(c_i) \leq \bar{\epsilon}T. \tag{17}$$

This implies that

$$\frac{1}{N} \sum_i^N [(J(\pi_i, c^*) - J(\pi_E, c^*)) - (J(\pi_i, c_i) - J(\pi_E, c_i))] \leq \bar{\epsilon}T, \tag{18}$$

as $\hat{c}$ is the best-in-hindsight constraint. We then note that $J(\pi_i, c_i) - J(\pi_E, c_i) \leq 0$ by the fact that $\pi_i$ is produced via a CRL procedure, which means we can drop the former term from the above sum, giving us

$$\frac{1}{N} \sum_i^N (J(\pi_E, c^*) - J(\pi_i, c^*)) \leq \bar{\epsilon}T. \tag{19}$$

Because this equation holds on average, there must be at least one $\pi \in \pi_{1:N}$ for which it holds. Now, we recall that $\pi_i = \text{CRL}(r, c_i)$ to complete the proof. $\square$

## A.2 Proof of Lemma 3.3

*Proof.* For a single $c$, a standard Hoeffding bound tells us that

$$P(|\frac{1}{K} \sum_{i=0}^K V_k(c) - \mathbb{E}[V(c)]| \geq \epsilon) \leq 2 \exp\left(\frac{-2K\epsilon^2}{(2T)^2}\right), \tag{20}$$

where $V_k(c)$ denotes the value of the payoff using data from the $k$th task. We have $|\mathcal{F}_c|$ constraints and want to be within $\epsilon$ of the population mean uniformly. We can apply a union bound to tell us that w.p. at least

$$1 - 2|\mathcal{F}_c| \exp(\frac{-2K\epsilon^2}{(2T)^2}), \tag{21}$$

we will do so. If we want to satisfy this condition with probability at least $1 - \delta$, simple algebra tells us that we must draw

$$K \geq O\left(\log\left(\frac{|\mathcal{F}_c|}{\delta}\right) \frac{(2T)^2}{\epsilon^2}\right) \tag{22}$$

samples. $\square$

## A.3 Proof of Theorem 3.5

*Proof.* For each $c \in \mathcal{F}_c$, define the set of safe policies as $\Pi(c) = \{\pi \in \Pi | J(\pi, c) \leq 0\}$. This set is non-empty by assumption. Define

$$u(\tau, c) = \mathbb{1}\{\pi_E^\tau \in \Pi(c)\}. \tag{23}$$

We prove each of the desired conditions independently.

**Reward Condition.** Let $\hat{c} \in c_{1:N}$. Recall that we want to prove that

$$\mathbb{E}_{\tau \sim P(\tau)}[J(\pi(r^\tau), r^\tau) - J(\pi_E^\tau, r^\tau)] \geq -2\epsilon T, \tag{24}$$

where $\pi^\tau = \text{CRL}(r^\tau, \hat{c}, \delta = 0)$. Observe that

$$\pi_E^\tau \in \Pi(\hat{c}) \Rightarrow J(\pi^\tau, r^\tau) = \max_{\pi \in \Pi(\hat{c})} J(\pi, r^\tau) \geq J(\pi_E^\tau, r^\tau). \tag{25}$$

Thus, if

$$\mathbb{E}_\tau[u(\tau, \hat{c})] \geq 1 - \epsilon, \tag{26}$$

we have that

$$\mathbb{E}_\tau[J(\pi^\tau, r^\tau) - J(\pi_E^\tau, r^\tau)] \tag{27}$$
$$\leq \mathbb{E}_\tau[(1 - u(\tau, \hat{c}))(J(\pi^\tau, r^\tau) - J(\pi_E^\tau, r^\tau))] \tag{28}$$
$$\leq \mathbb{E}_\tau[(1 - u(\tau, \hat{c}))](\sup_\tau J(\pi^\tau, r^\tau) - J(\pi_E^\tau, r^\tau)) \tag{29}$$
$$\leq \mathbb{E}_\tau[(1 - u(\tau, \hat{c}))]2T \tag{30}$$
$$\leq -2\epsilon T \tag{31}$$

We now prove that with $K$ large enough, we can guarantee Eq. 26 holds true w.h.p. Define

$$\widetilde{\mathcal{F}}_c = \{c \in \mathcal{F}_c | \forall k \in [K], \pi_E^K \in \Pi(\hat{c})\}. \tag{32}$$

We now argue that if

$$K \geq O\left(\log \frac{|\mathcal{F}_c|}{\delta} \frac{1}{\epsilon^2}\right), \tag{33}$$

w.p. $\geq 1 - \delta$, Eq. 26 holds true $\forall c \in \widetilde{\mathcal{F}}_c$. This means that as long as we pick $c_{1:N} \in \widetilde{\mathcal{F}}_c$, our desired condition will be true. Note that this is fewer than the number of samples we assumed in the theorem statement.

For a single constraint, a Hoeffding bound tells us that

$$P(|\frac{1}{K} \sum_k^K \mathbb{1}\{\pi_E^k \in \Pi(c)\} - \mathbb{E}_\tau[u(\tau, c)]| \geq \epsilon) \leq 2\exp\left(-2K\epsilon^2\right). \tag{34}$$

Union bounding across $\mathcal{F}_c \supseteq \widetilde{\mathcal{F}}_c$, we get that the probability that $\exists c \in \widetilde{\mathcal{F}}_c$ s.t. Eq. 26 does *not* hold is upper bounded by

$$1 - 2|\mathcal{F}_c| \exp\left(-2K\epsilon^2\right). \tag{35}$$

To have this quantity be $\geq 1 - \delta$, we need

$$K \geq O\left(\log \frac{|\mathcal{F}_c|}{\delta} \frac{1}{\epsilon^2}\right). \tag{36}$$

**Safety Condition.** We begin by considering the infinite sample setting. We therefore desire to prove that

$$\mathbb{E}_\tau[J(\pi^\tau, c^*) - J(\pi_E^\tau, c^*)] \leq \bar{\epsilon}T. \tag{37}$$

Define the per-round loss of the constraint player as

$$\ell_i(c) = \frac{1}{T} \mathbb{E}_\tau[J(\pi_i^\tau, c) - J(\pi_E^\tau, c)] \in [-1, 1], \tag{38}$$

the best-in-hindsight comparator as $\hat{c} = \text{argmax}_{c \in \mathcal{F}_c} \sum_i^N \ell_i(c)$, instantaneous regret as $\epsilon_i = \ell_i(\hat{c}) - \ell_i(c_i)$, and average regret as $\bar{\epsilon} = \frac{1}{N} \sum_i^N \epsilon_i$. Proceeding similarly to the single-task case,

$$\bar{\epsilon}T = \frac{1}{N} \sum_i^N \mathbb{E}_\tau[J(\pi_i^\tau, \hat{c}) - J(\pi_E^\tau, \hat{c})] - \mathbb{E}_\tau[J(\pi_i^\tau, c_i) - J(\pi_E^\tau, c_i)] \tag{39}$$

$$\geq \frac{1}{N} \sum_i^N \mathbb{E}_\tau[J(\pi_i^\tau, c^*) - J(\pi_E^\tau, c^*)] - \mathbb{E}_\tau[J(\pi_i^\tau, c_i) - J(\pi_E^\tau, c_i)] \tag{40}$$

We now argue that the second term in the above sum must be non-positive. Consider an arbitrary task $\tau$. Then, because $\pi_E^\tau \in \Pi(c_i)$ and CRL is optimizing over $\Pi(c_i)$, this term must be negative. As it is

negative per-task, it must be negative in expectation. Thus, we are free to drop the second term in the above expression which tells us that

$$\bar{\epsilon}T \geq \frac{1}{N}\sum_i^N \mathbb{E}_\tau[J(\pi_i^\tau, c^*) - J(\pi_E^\tau, c^*)] \tag{41}$$

Because this equation holds on average, there must be at least one $\pi \in \pi_{1:N}$ for which it holds. Now, we recall that $\pi_i^\tau = \text{CRL}(r^\tau, c_i)$ to complete the infinite-sample proof.

We now consider the error induced by only observing a finite set of tasks. There are two places finite-sample error enters: in estimating the value of $\ell_i(c)$ and in estimating $\widetilde{\mathcal{F}}_c$.

By Lemma 3.3, the maximum error we can induce by estimating $\ell_i$ from finite samples is upper bounded w.h.p by $\epsilon$. Thus, the extra error induced on the average regret is also bounded by $\epsilon$. Observe that our losses are scaled by $\frac{1}{T}$ in comparison to difference of $J$s. Therefore, we need to add an $\epsilon T$ to our bound for the infinite-sample setting.

By our argument in the reward section, $\pi_E^\tau \notin \Pi(c_i)$ w.p. $\leq \epsilon$. When this is true, $J(\pi_i^\tau, c_i) - J(\pi_E^\tau, c_i)$ can be as big as $2T$. Thus, the term we dropped in Eq. 40 ($V(c_i)$) can be as big as $2\epsilon T$ instead of 0. In the worst case, this adds an additional $2\epsilon T$ to our bound.

Combining both of the above, when we transition from the infinite sample setting to the finite sample setting, our bound degrades by $3\epsilon T$. $\qquad\square$

# B Experimental Details

An overview of the key experimental details is presented below. The exact code for reproducing our experiments is available at https://github.com/konwook/mticl.

## B.1 CRL

We use the Tianshou [Weng et al., 2022] implementation of PPO [Schulman et al., 2017] as our baseline policy optimizer. Classical Lagrangian methods exactly follow the gradient update shown in Algorithm 1, but they are susceptible to oscillating learning dynamics and constraint-violating behavior during training. The PID Lagrangian method [Stooke et al., 2020] extends the naive gradient update of $\lambda_i$ with a proportional and derivative term to dampen oscillations and prevent cost overshooting.

We find that augmenting the policy state with the raw value of the constraint function to be crucial for successful policy training. For CRL, this corresponds to the violation of the ground-truth constraint.

## B.2 Expert Demonstrations

For all of our experiments, our experts are CRL policies trained to satisfy a ground truth constraint. The expert demonstrations we use in ICL are generated by rolling out these policies and collecting 20 trajectories. To simulate suboptimal experts, we generate noisy trajectories by adding zero-mean Gaussian noise (with standard deviation 0.7 / 0.5 for velocity / position) to the policy's action at every timestep.

## B.3 Single-Task ICL

Similar to CRL, we augment the policy state with the raw value of the constraint function. For ICL, this corresponds to the violation of the learned constraint.

To reduce the amount of interaction required to solve the inner optimization problem, we warm-start our policy in each iteration by behavior cloning against the given expert demonstrations. We zero-out the augmented portion of the state when behavior cloning to avoid leaking constraint violation information from the expert.

When using CRL as part of ICL, we set the constraint threshold used in the Lagrangian update to be the expert's constraint violation. However, when starting with degenerate constraints, this can prevent policy optimization from learning at all as the expert's violation under the constraint can be arbitrarily low. To circumvent this issue, we set the Lagrangian constraint threshold to use the expert's violation plus a cost limit buffer, which we anneal over the course of training to 0. This ensures that our learned policy satisfies the learned constraint as much as the expert does as desired.

Because ICL requires learning a constraint, we represent our constraints as neural networks, mapping from the state space of our agent to a bounded scalar in the range $[0, 1]$. To update this constraint, we solve the optimization problem using a regression objective. Learner and expert constraint values are labeled with 1 and -1 respectively, and we optimize a mean squared error loss. We found that weighting data equally (instead of via returned Lagrange multipliers) was sufficient for good performance and therefore utilize this simpler strategy.

For both CRL and ICL, we find that using a log-activation on top of the raw value of the constraint is an important detail for stable training.

## B.4 Multi-Task ICL

For the multi-task maze setting, we consider 10 distinct tasks corresponding to unique goal locations, each with 2 starting locations.

Our solver combines a waypoint planner with a low-level controller. Waypoints are calculated by discretizing the maze into a 10 by 10 grid and running Q-value iteration, while low-level actions are computed using CRL experts trained to move in a single cardinal direction. To generate a trajectory, we follow a sequence of waypoints from the planner using the controller.

For ICL, we visualize our learned constraints by uniformly sampling 100,000 points from every grid cell, averaging the constraint predictions, and thresholding at a limit of 0.5.

## B.5 Hyperparameters

The key experimental hyperparameters are shown in Table 1. The exact configuration we use for our experiments is available at https://github.com/konwook/mticl/blob/main/mticl/utils/config.py.

| Hyperparameter | Value |
| --- | --- |
| PPO Learning Rate | 0.0003 |
| PPO Value Loss Weight | 0.25 |
| PPO Epsilon Clip | 0.2 |
| PPO GAE Lambda | 0.97 |
| PPO Discount Factor | 0.99 |
| PPO Batch Size | 512 |
| PPO Hidden Sizes | [128, 128] |
| P-update Learning Rate | 0.05 |
| I-update Learning Rate | 0.0005 |
| D-update Learning Rate | 0.1 |
| Constraint Batch Size | 4096 |
| Constraint Learning Rate | 0.05 |
| Constraint Update Steps | 250 |
| Steps per Epoch | 20000 |
| CRL Velocity Epochs | 50 |
| CRL Position Epochs | 100 |
| ICL Expert Demonstrations | 20 |
| ICL Velocity Cost Limit | 20 |
| ICL Position Cost Limit | 100 |
| ICL Anneal Rate | 10 |
| ICL Velocity Outer Epochs | 20 |
| ICL Position Outer Epochs | 10 |
| ICL Velocity Epochs | 10 |
| ICL Position Epochs | 50 |

Table 1: Experiment hyperparameters.

# C Comparison to Chou et al. [2020]

## C.1 Experimental Setup

We attempt to faithfully implement the method of Chou et al. on the problems we consider as a baseline for single-task ICL. At a high level, their method requires 2 steps:

1. For each expert trajectory, perform a random search in trajectory space starting from the demo to try and compute the set of trajectories that are higher reward than the demo.

2. Solve a constrained optimization problem over a known parametric family that labels each expert trajectory with +1 and each learner trajectory with 0.

Note that in contrast to our method, Chou et al. perform a single constraint estimation, rather than an iterative procedure.

For Step 1), Chou et al. [2020] use hit-and-run sampling that ensures good coverage over this set of trajectories in the limit of infinite sampling. However, for tasks of the horizon (1000+) and state-space dimension (30) we consider, one would need a rather large number of samples to ensure good coverage (i.e. an $\epsilon$-net would require $\frac{30^{1000}}{\epsilon}$ samples).

As search in the space of trajectories is infeasible, we instead search in the space of policies, as is much more standard on long-horizon problems. We therefore initialize the policy with behavioral cloning (similar to starting from the expert trajectories), performing maximum entropy RL (a search procedure with good coverage properties), and return the set of trajectories from the replay buffer that are of a higher reward than that of the expert.

For Step 2), Chou et al. [2020] use a variety of mixed-integer LP solvers. In essence however, they are trying to solve a classification problem. Of course, with their assumption of an optimal expert (or a "boundedly suboptimal" expert with a known sub-optimality gap), they can solve the strict feasibility problem. Because we assume that the expert is safe in expectation (rather than uniformly), it is natural to consider a relaxation of this problem (i.e. a convex relaxation of the 0/1 loss).

We experimented with several such relaxations when developing our method and found the $\ell_2$ relaxation (i.e. treating the problem as a regression problem with different targets for learner and expert demos) to work the best in practice. We therefore use the same relaxation for their method. For our single-task experiments, it is clear how to specify the "parametric family" the constraint fits in (e.g. a linear threshold), so we believe the above is a faithful implementation.

So, in short, we perform the same constraint learning procedure over a different source of learner data.

## C.2 Baseline Comparison Results

We implemented the above method on our velocity and position tasks and display the results (averaged over 3 seeds) below.

| Algo. | $|c^* - c|$ ($\downarrow$) | $J(\pi_E, r) - J(\pi, r)$ ($\downarrow$) | $J(\pi_E, c^*) - J(\pi, c^*)$ ($\uparrow$) |
|---|---|---|---|
| ICL (iter 1) | 0.052 | 569.277 | **270.111** |
| ICL (iter 10) | **0.000** | **-72.149** | 133.59 |
| Chou et al. | 0.311 | 384.772 | -1958.372 |

Table 2: Baseline Comparison (Position Constraint)

| Algo. | $|c^* - c|$ ($\downarrow$) | $J(\pi_E, r) - J(\pi, r)$ ($\downarrow$) | $J(\pi_E, c^*) - J(\pi, c^*)$ ($\uparrow$) |
|---|---|---|---|
| ICL (iter 1) | 0.249 | 12.267 | -244.25 |
| ICL (iter 10) | **0.006** | **-46.771** | **-21.602** |
| Chou et al. | 0.246 | -28.315 | -134.58 |

Table 3: Baseline Comparison (Velocity Constraint)

We see that our method performs better in terms of all 3 performance criteria we evaluate under. As implemented, Chou et al.'s method appears to produce an inaccurate constraint that leads to a significant safety violation (the bottom right cell in both tables).

