# OpenReview forum: "Learning Shared Safety Constraints from Multi-task Demonstrations"
_NeurIPS.cc/2023/Conference — NeurIPS 2023 poster_

### Official Review · Reviewer_Ray8 · 2023-06-28

**Soundness:** 4 excellent
**Presentation:** 3 good
**Contribution:** 3 good
**Rating:** 7
**Confidence:** 3

**Summary:**

The authors propose a novel method for constraint learning from expert demonstrations by an optimal CRL policy. The idea behind the algorithm is to frame the problem as a zero-sum game between a policy player and a constraint player, using inspiration from a two-player zero-sum game expression of IRL and CRL. With the reward function available, expert demonstrations, and a class of potential constraints containing the ground truth constraint, the authors propose to minimize the cost to go subject to being at least as safe as the expert, i.e. to obtain a larger cost than the expert policy on all constraints in the set.

The authors derive no regret convergence bounds for the learned constraints and deal with degeneracy issues by extending the problem to the multi-task framework with asymptotic guarantees. They present simulation results on high dimensional robotics tasks.

**Strengths:**

Very sound paper, and very clean and ingenuous framing of the problem as a game. I really enjoyed reading the paper as the work elegantly composes concepts from IRL, CRL, GT and no regret learning. I also see it as being of big significance for a very important problem, that of learning constraints from demonstrations. Overall very solid work.

**Weaknesses:**

- Although the algorithmic contribution is very strong, it seems that the practical algorithm section of the paper is much weaker and feels rushed. The single task results seem very simplistic while the multi-task maze example is way too briefly discussed. There is one sentence describing the threshold after which correct recovery of the constraints occurs, an aspect that would deserve a more extensive presentation and analysis.
- Along that line, figure 5 deserve to be reworked for clarity, it mixes too many things in together with little readability.
- There is no discussion on the computation costs of running the algorithm and on its data requirements. The authors only mention succinctly that CRL is slower than RL (by how much?), but it would be useful to know how many demonstrations one needs for typical robotics scenarios and how much compute the recovery of constraints needs.


**Questions:**

- How do the authors expect their solution to scale for systems where the state is much higher in dimension than the simple examples presented. For example do they think that their method could be a direction to learn constraints from sequences of images in applications such as autonomous driving or flight?

- (Optional) out of curiosity, I am wondering if the authors have any insight on how much more challenging it would be to consider constraints in multi-agent environments?

**Limitations:**

- The authors mention the lack of real world experiments as a limitation, I could not agree more but would go even further in suggesting that the proposed simulations are insufficient to help readers fully grasp the potential of the method. The paper would warrant a higher rating should that aspect be improved.

- I am awaiting the authors response to better understand whether the method would actually be practical to use for real life applications and robotics problems of interest.

---

> ### Author Rebuttal · Authors · 2023-08-10
>
> We thank the reviewer for their kind words about our work. Responding in order to the concerns raised:
>
> W1: Because of the overly conservative constraints single-task ICL produces on some problems (e.g. a maze), our main goal with the single-task experiments was to show that this is not always the case, if one is able to restrict the class of constraints appropriately. Specifically, on both of the high-dimensional continuous control problems we consider, we are able to recover the ground-truth constraint. In response to your concern about the multi-task experiments, as we mention in the global response, we performed experiments on a more complex problem: recovering the walls of the ant-maze (rather than the point maze).
>
> W2: We agree and would be more than happy to split up Figure 5 into multiple plots, given an additional page of space.
>
> W3: While our sample complexity analysis does provide an answer to the question of how many tasks we might need to see to recover a good constraint, we agree that we could add more details about how many demonstrations we used for our practical experiments. For all experiments, we used 20 demos from the expert per task. Once we’ve optimized the policy, learning a constraint is just a classification problem and is therefore of limited computational expense. The computational complexity of constrained RL vs. regular RL depends a lot on how related the constraint and the reward function are to each other (as if they are highly correlated, we are likely to need more Lagrange multiplier updates / policy optimization steps). It is therefore difficult for us to make a global statement.
>
> Q1: We would expect that learning constraints in a high-dimensional input space like images would be fundamentally difficult (for all methods) as different views of the same scene could correspond to the same level of constraint violation. However, for problems like self-driving, it is common to perform policy learning on top of a fixed multi-modal scene encoder rather than end-to-end learning. We would expect our approaches to scale quite well to these relatively low-dimensional representations.
> Q2: In theory, it shouldn’t be any more challenging to have multi-agent environments. We were actually thinking of adding in some experiments of this flavor but, because solving multi-agent RL problems can be challenging in and of itself, chose not to as we did not want to introduce additional complexities.
>
> L 1/2: We believe our method could be easily applied on top of a pre-existing planning stack. One direction we are currently working with some collaborators on is the problem of learning costmaps for off-road driving. There, one could imagine using our method to, by comparing human drivers and the output of a standard planner, learn the set of obstacles in the scene the ATV should avoid. Of course, anything with real robots takes more than a week to implement, but we do believe our method is applicable to these sorts of problems.

---

> > ### Comment · Reviewer_Ray8 · 2023-08-15
> > **Response to authors**
> >
> > I thank the authors for their clarifications. I have no other questions and maintain my review score.

---

### Official Review · Reviewer_doVy · 2023-07-05

**Soundness:** 3 good
**Presentation:** 2 fair
**Contribution:** 2 fair
**Rating:** 7
**Confidence:** 3

**Summary:**

The paper proposes a novel inverse constraint learning (ICL) approach that leverages constrained reinforcement learning (CRL) and a game-theoretic formulation. Specifically, the paper shows that the game-theoretic view of inverse reinforcement learning (IRL) naturally extends to ICL, by forming the Lagrangian of CRL. The resulting single-task ICL algorithm is shown to Pareto-dominate the expert policy in terms of its expected cumulative reward and the constraint violation. However, the recovered constraint may be too conservative and generalize poorly under a new task. To address this issue, the framework is extended to incorporate expert demonstrations under multiple tasks that share the same constraint.The paper shows that the multi-task ICL framework is expected to approximately Pareto-dominate the expert policies. Simulation results suggest the efficacy of the approach in continuous and discrete MDPs.

**Strengths:**

* Unlike prior ICL methods, the proposed framework is built on the game-theoretic approach. This brings in multiple benefits: 1) the form of the reward function can remain general; 2) conditions on approximate Pareto dominance can be derived, which gives theoretical justification of the proposed approach; 3) the resulting algorithm is relatively straightforward to implement.

* Figures 1 and 2 give a nice visual representation of the proposed approach. Algorithm 1, 2, and 3 are presented in a consistent and organized way that clarifies the differences among CRL, single-task ICL, and multi-task ICL.

**Weaknesses:**

* The most concerning point is that the paper provides no comparison to prior work in the experiment. Some important insights of the proposed approach is shared by the seminal work of [1]. Specifically, both approaches 1) observe that access to reward functions enables constraint learning by looking at the differences between safe demonstrations and unsafe optimal demonstrations; 2) can leverage safe demonstrations from multiple tasks. To strengthen the contribution of the paper, it should include qualitative and quantitative comparison to [1] in Section 4 (where applicable), to clarify the practical advantages of the proposed framework. This might involve evaluating the ICL algorithm in more realistic problems than the toy task of maze navigation, since the authors claim better scalability of their approach compared to [1] in realistic problems (in Section 2). In addition, the paper should give a more detailed description of [1] to better highlight the similarities and differences.

* There are multiple places in the paper where a variable or an acronym is used without definition: 1) $c^*$ in Theorem 3.1; 2) “FTRL” in line 153. Also, some figures in Section 4 are confusing and hard to interpret. Specifically, I believe that the blue cells in Figure 5(b) and (c) represent start and goal cells. However, the same color is used in other sub-figures to represent the constraints. It is recommended to use different colors to represent the start and the goals so that readers can make sense of Figure 5 as a whole. For further points concerning the presentation of the paper, please see the questions below.

[1] Chou, Glen, Dmitry Berenson, and Necmiye Ozay. "Learning constraints from demonstrations with grid and parametric representations." The International Journal of Robotics Research 40, no. 10-11 (2021): 1255-1283.

**Questions:**

* What is $c_i^k$ in Algorithm 3? Shouldn’t it be just $c_i$ since the constraint is shared across all the $K$ tasks?

* The order of minimization and maximization is swapped between (6) and (7). Does the equality hold because of the standard minimax theorem in game theory? If so, are there any conditions imposed on the form of $J$ in order for the equality to hold?

**Limitations:**

* The paper assumes access to demonstrations of the optimal safe policy for a given task. While the availability of safe demonstrations is a reasonable assumption,optimal demonstrations may be too demanding to ask for a human demonstrator in real-world settings [2]. It would be quite interesting to see how the proposed ICL framework handles suboptimal demonstrations; in theory, the objective function of equation (4) does not seem to require the optimality of the expert under the given reward function $r$.

[2] Xu, Haoran, Xianyuan Zhan, Honglei Yin, and Huiling Qin. "Discriminator-weighted offline imitation learning from suboptimal demonstrations." In International Conference on Machine Learning, pp. 24725-24742. PMLR, 2022.

---

> ### Author Rebuttal · Authors · 2023-08-10
>
> We appreciate that the reviewer found our figures / algorithm descriptions clear and that our algorithm seemed straightforward to implement – we think this bodes well for its application to real-world problems. Responding to the concerts raised:
>
>
> W1: Please see our global response for the relationship between our work and that of Chou et al. We would be more than happy to add the above discussion to our work (perhaps in the appendix as it is rather lengthy?). Please also see our global response on experimental complexity.
>
> W2: $c^*$ is the ground-truth constraint, as defined on line 127 – that being said, we would be happy to add a reminder of this point to the theorem statement. FTRL stands for follow the regularized leader, a standard no-regret algorithm – we apologize for not explaining this acronym and would be happy to add in a description. While we did label the start/goal and maze squares differently, we can see why their shared color would be confusing and would be happy to use a different color.
>
> Q1: You are totally right, we will update that in our next draft.
>
> Q2: Yes, this is allowed because of the minimax theorem. In the setup of the problem, we assumed the conditions for Sion’s minimax theorem to hold (convex / compact strategy spaces). Even more simply, in the finite policy / constraint class setting, we are solving a matrix game for which von Neumann’s original minimax theorem holds.
>
> L1: This is an excellent point we didn’t fully appreciate until the reviewer pointed it out – thanks! We don’t actually require the expert to be the optimal policy, only that they are as safe as we want our agent to be. Our Thms. 3.1 / 3.5 clearly allows for the learner to return a policy that out-performs that of the expert. We would be more than happy to re-word any claims to the contrary in our current draft. Please see our global response for experimental evidence that our method is able to take advantage of suboptimal demonstrations.

---

> > ### Author Response · Authors · 2023-08-17
> >
> > We thank the reviewer for their thoughtful comments. As we get closer to the end of the discussion period, please let us know if we were able to answer all of your questions or if there is anything else we could clarify.

---

> > > ### Comment · Reviewer_doVy · 2023-08-20
> > > **Acknowledgement of Author Response**
> > >
> > > Dear authors,
> > > I apologize for my delayed response. Thank you for providing detailed explanations, especially around the differences between your work and the work of Chou et al. I would highly recommend that you add the discussion in the body of the paper, at least in the Appendix (and summarize the key points in the main paper as much as possible). Thank you also for conducting the experiments with suboptimal experts. It is indeed strong empirical evidence that can differentiate your work from the majority of prior work that requires a soft-optimal expert. As a result, I acknowledge the author response and am going to raise my score. However, I still believe that a baseline comparison is highly desirable for a more objective evaluation, at least in the simpler single-task setup. (For instance, it seems that a linear position constraint in Figure 4 could be learned via parametric or grid-based constraint representation that some prior work can handle.)

---

> > > > ### Author Response · Authors · 2023-08-21
> > > > **Re:**
> > > >
> > > > Hi,
> > > >
> > > > In response to your request and that of another reviewer, we spent some time thinking about how we could reasonably faithfully implement the method of Chou et al. on the problems we consider. At a high level, their method requires 2 steps:
> > > >
> > > > 1) For each expert trajectory, perform a random search in trajectory space starting from the demo to try and compute the set of trajectories that are higher reward than the demo.
> > > > 2) Solve a constrained optimization problem over a known parametric family that labels each expert trajectory with +1 and each learner trajectory with 0.
> > > >
> > > > Note that in contrast to our method, Chou et al. perform a *single* constraint estimation, rather than an iterative procedure.
> > > >
> > > > For 1), they use hit-and-run sampling that ensures good coverage over this set of trajectories in the limit of infinite sampling. However, for tasks of the horizon (1000+) and state-space dimension (~30) we consider, it seems like one would need an astronomical number of iterations to ensure good coverage (i.e. an $\epsilon$-net would require $\frac{30^{1000}}{\epsilon}$ samples). This is part of the reason hit-and-run sampling is not used in practice for solving even regular Mujoco tasks (and, to the best of our knowledge, is mostly applied on bandit problems). While we weren't able to find descriptions of the horizons of the task Chou et al. consider, we would be rather surprised if they were in the thousands. If search in the space of trajectories is infeasible, we could instead try search in the space of *policies*, as is much more standard on long-horizon problems. So, we propose initializing a policy with behavioral cloning (similar to starting from the expert trajectories), performing maximum entropy RL (a search procedure with good coverage properties), and returning the set of trajectories from the replay buffer that are of a higher reward than that of the expert.
> > > >
> > > > For 2), they use a variety of mixed-integer LP solvers. In essence however, they are trying to solve a classification problem. Of course, with their assumption of an optimal expert (or a "boundedly suboptimal" expert with a known sub-optimality gap), they can solve the strict feasibility problem. Because we assume that the expert is safe in expectation (rather than uniformly), it is natural to consider a relaxation of this problem (i.e. a convex relaxation of the 0/1 loss). We experimented with several such relaxations when developing our method and found the $\ell_2$ relaxation (i.e. treating the problem as a regression problem with different targets for learner and expert demos) to work the best in practice. It therefore seemed the most fair for us to use the same relaxation for their method. As a reviewer pointed out, for our single-task experiments, it is clear how to specify the "parametric family" the constraint fits in (e.g. a linear threshold), so we believe the above is a faithful implementation.
> > > >
> > > > So, in short, we perform the same constraint learning procedure over a different source of learner data. We implemented the above method on our position task and display the results below:
> > > >
> > > > | Algo.         | $\|c^* - c\|$ (↓) | $J(\pi_E, r) - J(\pi, r)$ (↓) | $J(\pi_E, c^*) - J(\pi, c^*)$ (↑) |
> > > > |---------------|-----------------|-----------------------------|---------------------------------|
> > > > | ICL (iter 1)  | 0.052           | 569.277                     | **270.111**                     |
> > > > | ICL (iter 10) | **0.000**       | **-72.149**                 | 133.59                          |
> > > > | Chou et al.   | 0.311           | 384.772                     | -1958.372                       |
> > > >
> > > > We see that our method performs better in terms of all 3 performance criteria we evaluate under. As implemented, Chou et al.'s method appears to produce an inaccurate constraint that leads to a significant safety violation (the bottom right cell in the above table), even when compared to the first iteration of our iterative procedure.
> > > >
> > > > We hope these experiments help resolve any remaining concerns you might have. Please let us know if you have any other questions.

---

> > > > > ### Comment · Reviewer_doVy · 2023-08-21
> > > > > **Thank you for the additional experiment**
> > > > >
> > > > > Dear Authors,
> > > > >
> > > > > Thank you for conducting the comparative study and providing the details of the experimental setup. The design of the experiment seems reasonable to me, and the empirical results clearly exhibit superiority of ICL over the prior work. I am going to raise the score once again as all my concerns so far have been addressed. Thank you for your hard work despite the short period of time.

---

> > > > > > ### Author Response · Authors · 2023-08-21
> > > > > > **Re:**
> > > > > >
> > > > > > Thank you -- we appreciate your continued engagement! For completeness, we also include below the result for velocity constraints:
> > > > > >
> > > > > > | Algo.         | $\|c^* - c\|$ (↓) | $J(\pi_E, r) - J(\pi, r)$ (↓) | $J(\pi_E, c^*) - J(\pi, c^*)$ (↑) |
> > > > > > |---------------|-----------------|-----------------------------|---------------------------------|
> > > > > > | ICL (iter 1)  | 0.249           | 12.267                      | -244.25                         |
> > > > > > | ICL (iter 20) | **0.006**       | **-46.771**                 | **-21.602**                     |
> > > > > > | Chou et al.   | 0.246           | -28.315                     | -134.58                         |
> > > > > >
> > > > > > One again, our iterative method performs better than the baseline. Both this and the results in the previous table were averaged over three seeds. We would be more than happy to add these results to the paper.
> > > > > >
> > > > > > We also spent some time thinking about implementing Chou et al's method on our maze problem. Even if we ignore the difficulties with picking a parametric family that captures the set of all possible mazes, ignore that we were learning constraints on raw x / y positions rather than in the 10 x 10 grid space, and focus purely implementing step 1) on the grid (i.e. sampling shorter paths than the expert took), we would need to generate an number of trajectories that we are incapable of storing even a reasonable fraction of in memory. Specifically, we wrote a program to perform the dynamic programming calculations and it told us we would need to generate 7,126,295,908 trajectories to fully implement the multi-task version of their method. If we had to guess, this is why Chou et al. evaluate their method on smaller grids than we consider (as one pays exponentially in the height / width of the maze). We hope that the computational intractability of implementing their multi-task method combined with their poor performance on the simpler single-task problem is sufficient evidence of the superiority of our method.

---

### Official Review · Reviewer_3RKE · 2023-07-06

**Soundness:** 3 good
**Presentation:** 3 good
**Contribution:** 2 fair
**Rating:** 5
**Confidence:** 4

**Summary:**

This paper proposes to use inverse constraint learning (ICL) in multi-task scenarios. It uses game solving to describe the ICL problem, and then illustrates the learning algorithm under the single-task and multi-task problems. The authors give a theoretical analysis to explain under what conditions the learned reward can be generalized. The author shows the correctness of the proposed algorithm in both single-task and multi-task control experiments.

**Strengths:**

- The motivation of this paper is clear and reasonable to extract common safety-constraint from multi-task.
- The paper considers whether the learned reward and constraint can be generalized theoretically, and gives two conditions and related proofs.

**Weaknesses:**

- The paper did not give a comparison with the ICL algorithms mentioned in the related work, and did not show the advantages of the algorithm in both learning or safety concerns.
- The experimental settings of the paper is too simple, and only consider two constraints that do not require complex approachs to learn. It makes the machine learning method shows no advantages on human priors.
- The authors did not give the results of robotics control tasks (both in simulation and real-world) that really need safety-constraints, such as robotic manipulation with human.

**Questions:**

- Can the author supplement the experimental results on more complex robotics control tasks? For example, consider such a manipulation task as follow. We hope that the force or speed of the robot arm will not be too large while picking up an object. Can the algorithm learn such complex safety-constraints?

- The authors' experiment in multi-task scenarios essentially improves the diversity of samples through randomization. We generally don't think this is a real multi-task RL scenario because all tasks are navigation. Can the author explore diverse but related manipulation tasks (like push and pull an object) in the open-source taskset (such as [meta-world](https://meta-world.github.io/)) to show the effectiveness of the proposed method?

**Limitations:**

It is good that the paper focuses on the safety problems of reinforcement learning for robotics. However I suggest that the authors use more realistic robotics tasks (at least in simulation) that really need to consider safety-constraint to verify the method. Regardless of the comparison with related works or the complexity of the task, the experiments in the paper are not solid enough to support good motivation and theoretical results.

---

> ### Author Rebuttal · Authors · 2023-08-10
>
> We thank the reviewer for their appreciation of our theoretically-grounded approach. Responding to the concerns raised:
>
> W1: Please see our global response for the relationship between our work and that of Chou et al.
>
> W2: We’re not exactly sure what a “human prior” is here, could the reviewer clarify?
>
> W3/Q1/Q2: If we understand correctly, all three of these bullet points are in essence asking the same question: “what evidence do you have that the proposed method can scale to tasks like manipulation around people?
>
> **We would ask the reviewer to first read our global response on experiment complexity.**
>
> At a high level, our focus in this paper is not robotic manipulation specifically, but on learning with constraints more abstractly. Thus, we would argue that a lack of experiments in this particular domain is not a sufficient reason to reject our paper.
>
> As we point out in our global response, much of the prior work in inverse constraint learning focuses on tabular problems, while most of the work in constrained reinforcement learning focuses on relatively low dimensional tasks. So, we would argue that compared to the prior art, we are considering strictly harder settings.
>
> While we agree that focusing on manipulation specific applications would be interesting, our current experiments are equally high-dimensional (e.g. for our ant experiments, the state-space is 27 dimensional and the action space is 8 dimensional, compared to a 7 DoF manipulator) and we show results where are able to recover workspace constraints. More explicitly, given that we can learn a position constraint for a high-dimensional ant agent, we should also be able to learn constraints on the end effector of a manipulator). Similarly, if we are able to handle data that comes from the agent navigating to different parts of the maze, placing a block in different locations / goals should not be that different. Hence we see no conceptual reason why the algorithm would not scale to manipulation.

---

> > ### Comment · Area_Chair_V7pE · 2023-08-16
> > **Please clarify "human prior"**
> >
> > Dear Reviewer 3RKE,
> >
> > In order to give the authors an opportunity to address your W2, can you briefly respond to their question regarding a "human prior"? Thank you very much!

---

> > ### Comment · Reviewer_3RKE · 2023-08-17
> >
> > I believe that my concern is because I am lack of experience on this domain (safe RL). So till now I understand that the single-task experiment is showing that, if there is an unknown position or velocity constraint (defined by a linear function), the policy can learn not to break the constraint?
> >
> > The human prior that I have mentioned is that, the position and velocity constraints can be fully considered and easily learned with simple reward shaping. But if the safe RL area aims to set a task with the totally unknown constraints and only shows the training process can discover the constraints, I agree that it is too strict to evaluate this paper in a much more realistic view.
> >
> > I can raise my score due to that I did not understand the safe RL area, and the rebuttal has addressed my main concern. But I am also willing to see the future work on the real robotic tasks that really face to the safety problems.

---

> > > ### Author Response · Authors · 2023-08-17
> > > **Re:**
> > >
> > > Thanks! Yeah, the single-task experiments show that we are able to learn safe policies without knowledge of the ground-truth constraint.

---

### Official Review · Reviewer_NZBn · 2023-07-07

**Soundness:** 2 fair
**Presentation:** 2 fair
**Contribution:** 2 fair
**Rating:** 4
**Confidence:** 2

**Summary:**

Broadly, the paper address the challenge of safety constraints for robots, in particular the difficulty of handcrafting these. They propose learning safety constraints from expert demonstrations, particularly in a multi-task setting where each task reward is known and the safety constraints are task-agnostic.

They review related work on inverse RL (IRL), constrained RL (CRL), inverse constraint learning (ICL), and multi-task ICL. Compared to existing ICL approaches, they claim their approach is more general, provides more guarantees, and is simpler to implement.

They extensively formalize ICL, including explaining relevant approaches to IRL and CRL. As a brief recap, they compare ICL to IRL: IRL has an outer loop that searches over potential rewards, with an inner loop that uses RL to find an optimal policy. ICL has an outer loop that searches over potential constraints, with an inner loop that uses CRL to find a safe-optimal policy. In either case, the optimized policy is compared to the expert demonstrations to revise the learned reward or constraint.

They also explain a challenge with single-task ICL, namely that ICL may learn an overly-conservative safety constraint that simply forbids all states not visited by the expert demonstrations. This motivates their exploration of multi-task ICL, where the safety constraint is assumed to be shared among the tasks. Their primary novel contribution of the paper is an extension of a single-task ICL approach to multi-task: "we alternate between solving K CRL problems and updating the constraint based on the data from all policies". Using it, they prove that they can exactly identify a shared constraint given enough expert policies, under certain very strong assumptions. They also prove a weaker but more practical theorem: if they have enough tasks, they can learn a constraint that, if used to optimize a policy, leads to policies that don't exceed the constraint-violation of the expert demonstrations and do meet or exceed the reward of the expert demonstrations (within a tolerance).

For the single-task case, they evaluate their approach in the simulated ant environment in PyBullet and MuJoCo benchmarks. They choose arbitrary velocity and position constraints. They represent their learned constraints as neural networks, mapping from the state space of their agent to a bounded scalar in the range [0, 1]. They show that their approach can recover these ground-truth constraints, under certain simplifying assumptions (linear constraints; usage of a subset of the agent state as input).

For multi-task, they use the PointMaze environment from D4RL, with each task variant having one of two start positions and one of 10 goal positions. Using their approach, they are able to approximately learn the ground-truth constraints (namely the maze's blocked cells). They also show that naive approaches for combining 10 separate learned constraints from single-task ICL don't work well.

**Strengths:**

Their premise, that safety constraints are difficult to engineer and should be task-agnostic, is accurate and relevant to this domain.

Their approach is a straightforward extension of an existing single-task ICL approach to multi-task. It works well and matches the intuition that safety constraints should be task-agnostic.

Rigorous math in section 3 to formalize their work.

**Weaknesses:**

In section 3, it's not clear what is prior work and what is their novel contribution.

In 3.1, the authors copied several sentences verbatim from "Inverse Reinforcement Learning without Reinforcement Learning" (https://arxiv.org/pdf/2303.14623.pdf, section 3.1). The use of copied phrases like "we solve" suggest the authors, themselves, did this work.

In their single-task and multi-task results, they don't compare to any baselines, for example "Learning constraints from demonstrations" from Chou et al.

For readability, it would be helpful if they numbered references.

**Questions:**

Section 3.4 and 3.5 mention needing expert policies. However, the premise of your work is that you have access to expert demonstrations and reward functions, but not expert policies. Can you clarify?

A suggestion for positioning this line of work: instead of "safety constraints", I might argue that a more broad (and thus more valuable) set of constraints to be learned from human demonstrations is "task-agnostic human-behavior constraints", which would encompass safety plus other conventions and preferences. E.g. coming back to the example in your introduction, if I ask my friend to make toast, I'd be surprised if they set a plate down upside-down or rinsed the bread under the faucet.

>we consider a setting in which we have access to demonstrations of the optimal safe policy for a task, along with knowledge about the task’s reward

Can you describe this setting in practical terms? It seems that your approach has stronger requirements, namely (1) access to not just the rewards received by the demonstrations, but access to the reward function itself, and (2) access to a simulator or other mechanism to train a policy using CRL. As a clarifying example: if you had access to expert trajectories of a real-world robot including full scene state and reward annotations, across multiple tasks, this alone would not be sufficient to learn safety constraints using your method, right?

>In short, if we observe enough tasks, we are able to learn a constraint that, when optimized under, leads to policies that approximately Pareto-dominate that of the expert.

Can you elaborate? What is the specific Pareto-dominance here? My understanding is: you can learn a constraint that, if used to optimize a policy, leads to policies that don't exceed the constraint-violation of the expert demonstrations and do meet or exceed the reward of the expert demonstrations (within a tolerance). However, I don't intuitively follow why this should be your goal for the learned safety constraint.

For Figure 2, can you offer an intuitive explanation? For example:

= can you label the axes?

= why is an expert policy a single point in this space?

= what does the boundary of the polytope represent?

= what does this mean? "diversity of expert policies, none of which are optimal along the reward vector"

**Limitations:**

No concerns here

---

> ### Author Rebuttal · Authors · 2023-08-10
>
> We thank the reviewer for an incredibly detailed summary of our work which evinces a very thorough reading of our work.
>
> W1: Sec. 3.1 and 3.2 are standard algorithmic techniques. We would be happy to add something like “Prior Work” to the subsection titles.
>
> W2: If we understand correctly, the reviewer is referring to lines 102-104 and 111-115. These are standard sets of definitions (i.e. 102-104 is the definition of an MDP, 111-115 is the standard description of game-theoretic IRL). Given we are describing prior work in this section, we chose to stick to the notation / assumptions (e.g. convexity / compactness of strategy spaces) that is common in the literature. We also note that there are differences (e.g. we search over stationary policies) from the particular paper the reviewer mentions.
>
> W3. Our lack of comparisons to baselines in single-task ICL is because there are limited prior methods for this problem that scale to continuous control tasks (e.g. McPherson et al. assume we can perform value iteration on a tabular MDP). We compare to several baselines in the multi-task setting (e.g. the average / max of the learned constraints) that show that using multi-task data is critical for successful constraint learning. See the global response for the connection to Chou et al.
>
> W4: Sure!
>
> Q1:  We apologize for any confusion: our methods definitely do not require a DAgger-style interactive expert.  We believe the reviewer is referring to terms like those that appear in Equation (12). This is mostly a notational issue: given access to trajectories from some $\pi$, we can evaluate the value of the policy $J(\pi, r)$ for an arbitrary reward function $r$ without query-access to the policy by simply re-labeling the demonstrations with the reward function of interest . $\rho_{\pi_E}$, the visitation distribution of the expert, once again does not actually require query access to the expert policy, so we can evaluate the expression in Equation (13).
>
> Q2: This is an interesting point that gets at the difference between rewards and constraints. We usually think of a reward term as something with a fixed weight across tasks that gently shapes behavior. In contrast, a constraint is a term in your cost function that can be arbitrarily scaled up until the learner is no longer violating it. We would think that conventions that can be sometimes violated (e.g. setting the plate upside down to let the bottom dry off) to fit more under the purview of rewards. That being said, it is an interesting question as to whether our method would end up learning conventions.
>
> Q3: In the inverse reinforcement learning literature, access to demonstrations and the environment is standard. However, once we move to the space of constraints, we require an additional piece of information: the expert’s reward. Intuitively, the reason knowledge of the reward function is important is that it allows us to distinguish between an action not being taken because it was a) unsafe and b) suboptimal. Without this piece of information, we would have to assume that *all* untaken actions were unsafe and would therefore likely recover an overly conservative constraint. We note that *all* prior published work on ICL that we are aware of also makes this assumption.
>
> Q4: Your understanding of what we meant by “Pareto dominate” is entirely correct. When thinking about learning constraints, one could either a) attempt to recover the ground-truth constraint or b) try to learn a constraint that allows the learner to act safely / performantly. As we discuss in Sec. 3.5, the former can be an unreasonably challenging goal for realistic problems as it requires a high diversity and number of expert policies.
>
> However, this is not a problem unique to ICL: it also shows up in inverse reinforcement learning. In IRL, there can be multiple reward functions that make the expert policy optimal (e.g. a constant reward of zero). Thus, we try to learn a reward function where, if we computed the optimal policy under it, we would be guaranteed to have a bounded sub-optimality with respect to the expert policy under the ground-truth reward function. In essence, this is what the game-solving procedure guarantees us.
>
> In the ICL setting, we instead want to learn a constraint that explains the expert behavior (e.g. one that forbids highly rewarding but untaken behavior). The guarantees we get (that we learn a constraint that allows our learner to act safely / performantly) are analogous to the IRL guarantees and are in essence the strongest we could hope for without restrictive assumptions. This is part of what distinguishes our analysis from that of Chou et al., who focus on constraint recovery guarantees.
>
> Q5: We would be more than happy to add more description of this figure. At a high level, $\rho_{\Pi}$ refers to the space of occupancy measures (i.e. the set of state-action distributions of all policies in our policy class). The reason we chose this representation was that a constraint is simply a hyperplane in this space (the red line labeled $\langle \rho_{\pi}, c^* \rangle = 0$). To uniquely determine a line in $\mathbb{R}^d$, we need $d$ points on the line – these are what the green dots / expert policies $\pi_E^1$ and $\pi_E^2$ correspond to. We need two policies as our visitation distributions are two-dimensional (e.g. a two-state MDP). The technical conditions in Lemma 3.2 are the more formal version of the above statements (e.g. relint meaning the points are on the line rather than on the boundary of the space). Each corner of the polytope represents a policy, while the boundaries represent a convex combination of two policies. “Diversity of expert policies” means distinct policies and “not optimal along the reward vector” refers to their being a policy that performs better than the expert under the reward function (otherwise, we don’t need to solve a constrained problem at all and therefore the expert policy is not useful for extracting a constraint).

---

> > ### Comment · Reviewer_NZBn · 2023-08-16
> >
> > Thanks for your rebuttal and apologies for the lateness of my reply.
> >
> > >variety of baselines (e.g. the max, mean, or individual single-task constraints)
> >
> > These seem like weak baselines. It's not clear that they are "prior state of the art" or otherwise a well-regarded existing approach for recovering shared safety constraints from multiple tasks.
> >
> > Do you have any references to prior published work that utilize/advocate any of these approaches?
> >
> > I will continue reviewing your rebuttal and post a longer response later today.

---

> > > ### Author Response · Authors · 2023-08-16
> > > **Re:**
> > >
> > > Thanks, we look forward to your full response! Re: the multi-task baselines: to the best of our knowledge, there is only one other published work that considers the multi-task ICL problem, which is that of Chou et al. As we discussed in our global response, re-implementing their method requires access to several custom solvers that differ based on the particular family of constraints one wishes to search over, making it difficult for us to directly compare. We therefore turned out attention to single-task baselines, of which we compare to several reasonable ones.

---

> > ### Comment · Reviewer_NZBn · 2023-08-17
> >
> > >[Chou et al] did not release their code, it would be quite an undertaking for us to re-implement their methods on the problems we consider.
> >
> > As I'm sure you would agree, this (very) inconvenient fact doesn't exempt your work from comparison against strong baselines.
> >
> > >much of the prior work in inverse constraint learning focuses on tabular problems, while most of the work in constrained reinforcement learning focuses on relatively low dimensional tasks. So, we would argue that compared to the prior art, we are considering strictly harder settings.
> >
> > Unfortunately I don't have enough confidence in this domain to agree/disagree here.
> >
> > >It would be interesting to determine whether the proposed method can work for tasks beyond maze locomotion, which lack complex interactions with the environment.
> >
> > I agree with this reviewer's comment. In particular, the complexity I'd like to see is not just high dimensionality, but also long horizon length (e.g. avoiding a hot stove until it has cooled) and low predictability (e.g. keeping a larger distance from a pet or child that might move erratically).
> >
> > I'm not able to increase my rating. I suspect that a proper review of your work requires a deeper understanding of IRL, CRL, and ICL than I can offer. There's a lot of information in your rebuttal and reviewer discussions that I wasn't able to absorb, so I'm going to lower my confidence score.

---

> > > ### Author Response · Authors · 2023-08-17
> > >
> > > We thank the reviewer for their candor. We'd like to respond on a few points.
> > >
> > > 1. Beyond the fact that it is "very inconvenient" to implement the method of Chou et al., it is also extremely easy for us to construct problems for which our method readily applies but one cannot apply the method of Chou et al. For example, if we make one of the blocked sections in the maze a *circle* instead of a square, one can no longer use the "axis-aligned rectangles" technique of Chou et al. to recover a reasonable constraint. It would be rather shocking to us if a deep neural network could not fit this circle. Furthermore, Chou et al.'s method requires knowing the constraint family (e.g. "axis-aligned rectangles") a priori, while ours does not. This makes our techniques far more easily applicable to real-world problems where one does not usually know the exact parametric family of the constraint they are trying to fit, a limitation acknowledged by Chou et al.
> > >
> > > 2. The *minimum* horizon task we consider is of length 1000, while our maze task goes up to several thousand. So, we would argue that we are already learning constraints on long-horizon behavior.
> > >
> > > If there are any pieces of our above responses or our responses to other reviewers that could benefit from a more in-depth explanation, please let us know. We thank the reviewer for their engagement with our work!

---

> > > > ### Comment · Reviewer_NZBn · 2023-08-17
> > > >
> > > > >This makes our techniques far more easily applicable to real-world problems
> > > >
> > > > This is fine and a strength of your approach. But it sounds like there is a (restricted) problem you could construct where both approaches could apply, and then they could be compared.
> > > >
> > > > My earlier comment:
> > > > >In particular, the complexity I'd like to see is not just high dimensionality, but also long horizon length
> > > >
> > > > Your response above:
> > > > >The minimum horizon task we consider is of length 1000
> > > >
> > > > I was referring to the horizon length of the constraint, not the task. Consider learning a safety constraint around avoiding a hot stove. I'm assuming your state space includes the current on/off state of the stove (which we can imagine a smart home sensor or perception system providing), while your state space doesn't include the current stove-element temperature, as this is much harder to provide. Thus your approach needs to learn a safety constraint that is based on a memory of past environment state (how long ago did it get turned off? how hot did it get before it was turned off?).

---

> > > > > ### Author Response · Authors · 2023-08-17
> > > > > **Re:**
> > > > >
> > > > > Ah, thank-you for the clarification. While we are learning over-the-horizon constraints (i.e. a maximum cumulative cost over an episode), our method does not directly handle the sort of partially observed problem you describe above (i.e. a problem where the learner needs to perform some sort of filtering over history to extract the state required to represent the constraint). That being said, even in the safe RL setting (where we know the constraint a priori), it is not clear whether we can solve partially observed problems well without strong assumptions. This is because the information gathering actions the learner might need to take to reduce their uncertainty over the underlying state of the environment might end up violating the constraint (e.g. if the constraint was to never get close to a hot stove and the learner is only able to measure a temperature when they are close to an object). So, while we agree that this is an interesting question, it seems orthogonal to our paper, which focuses on constraint inference. However, we would be happy to clarify that we assume full observability in our paper.

---

> > > > > > ### Author Response · Authors · 2023-08-21
> > > > > > **Re:**
> > > > > >
> > > > > > Hi,
> > > > > >
> > > > > > In response to your request and that of another reviewer, we spent some time thinking about how we could reasonably faithfully implement the method of Chou et al. on the problems we consider. At a high level, their method requires 2 steps:
> > > > > >
> > > > > > 1) For each expert trajectory, perform a random search in trajectory space starting from the demo to try and compute the set of trajectories that are higher reward than the demo.
> > > > > > 2) Solve a constrained optimization problem over a known parametric family that labels each expert trajectory with +1 and each learner trajectory with 0.
> > > > > >
> > > > > > Note that in contrast to our method, Chou et al. perform a *single* constraint estimation, rather than an iterative procedure.
> > > > > >
> > > > > > For 1), they use hit-and-run sampling that ensures good coverage over this set of trajectories in the limit of infinite sampling. However, for tasks of the horizon (1000+) and state-space dimension (~30) we consider, it seems like one would need an astronomical number of iterations to ensure good coverage (i.e. an $\epsilon$-net would require $\frac{30^{1000}}{\epsilon}$ samples). This is part of the reason hit-and-run sampling is not used in practice for solving even regular Mujoco tasks (and, to the best of our knowledge, is mostly applied on bandit problems). While we weren't able to find descriptions of the horizons of the task Chou et al. consider, we would be rather surprised if they were in the thousands. If search in the space of trajectories is infeasible, we could instead try search in the space of *policies*, as is much more standard on long-horizon problems. So, we propose initializing a policy with behavioral cloning (similar to starting from the expert trajectories), performing maximum entropy RL (a search procedure with good coverage properties), and returning the set of trajectories from the replay buffer that are of a higher reward than that of the expert.
> > > > > >
> > > > > > For 2), they use a variety of mixed-integer LP solvers. In essence however, they are trying to solve a classification problem. Of course, with their assumption of an optimal expert (or a "boundedly suboptimal" expert with a known sub-optimality gap), they can solve the strict feasibility problem. Because we assume that the expert is safe in expectation (rather than uniformly), it is natural to consider a relaxation of this problem (i.e. a convex relaxation of the 0/1 loss). We experimented with several such relaxations when developing our method and found the $\ell_2$ relaxation (i.e. treating the problem as a regression problem with different targets for learner and expert demos) to work the best in practice. It therefore seemed the most fair for us to use the same relaxation for their method. As a reviewer pointed out, for our single-task experiments, it is clear how to specify the "parametric family" the constraint fits in (e.g. a linear threshold), so we believe the above is a faithful implementation.
> > > > > >
> > > > > > So, in short, we perform the same constraint learning procedure over a different source of learner data. We implemented the above method on our position task and display the results below:
> > > > > >
> > > > > > | Algo.         | $\|c^* - c\|$ (↓) | $J(\pi_E, r) - J(\pi, r)$ (↓) | $J(\pi_E, c^*) - J(\pi, c^*)$ (↑) |
> > > > > > |---------------|-----------------|-----------------------------|---------------------------------|
> > > > > > | ICL (iter 1)  | 0.052           | 569.277                     | **270.111**                     |
> > > > > > | ICL (iter 10) | **0.000**       | **-72.149**                 | 133.59                          |
> > > > > > | Chou et al.   | 0.311           | 384.772                     | -1958.372                       |
> > > > > >
> > > > > > We see that our method performs better in terms of all 3 performance criteria we evaluate under. As implemented, Chou et al.'s method appears to produce an inaccurate constraint that leads to a significant safety violation (the bottom right cell in the above table), even when compared to the first iteration of our iterative procedure.
> > > > > >
> > > > > > We hope these experiments help resolve any remaining concerns you might have. Please let us know if you have any other questions.

---

> > > > > > > ### Author Response · Authors · 2023-08-21
> > > > > > > **Re**
> > > > > > >
> > > > > > > We also provide the result for velocity constraints:
> > > > > > >
> > > > > > > | Algo.         | $\|c^* - c\|$ (↓) | $J(\pi_E, r) - J(\pi, r)$ (↓) | $J(\pi_E, c^*) - J(\pi, c^*)$ (↑) |
> > > > > > > |---------------|-----------------|-----------------------------|---------------------------------|
> > > > > > > | ICL (iter 1)  | 0.249           | 12.267                      | -244.25                         |
> > > > > > > | ICL (iter 20) | **0.006**       | **-46.771**                 | **-21.602**                     |
> > > > > > > | Chou et al.   | 0.246           | -28.315                     | -134.58                         |
> > > > > > >
> > > > > > > One again, our iterative method performs better than the baseline. Both this and the results in the previous table were averaged over three seeds. We would be more than happy to add these results to the paper.
> > > > > > >
> > > > > > > We also spent some time thinking about implementing Chou et al's method on our maze problem. Even if we ignore the difficulties with picking a parametric family that captures the set of all possible mazes, ignore that we were learning constraints on raw x / y positions rather than in the 10 x 10 grid space, and focus purely implementing step 1) on the grid (i.e. sampling shorter paths than the expert took), we would need to generate an number of trajectories that we are incapable of storing even a reasonable fraction of in memory. Specifically, we wrote a program to perform the dynamic programming calculations and it told us we would need to generate 7,126,295,908 trajectories to fully implement the multi-task version of their method. If we had to guess, this is why Chou et al. evaluate their method on smaller grids than we consider (as one pays exponentially in the height / width of the maze). We hope that the computational intractability of implementing their multi-task method combined with their poor performance on the simpler single-task problem is sufficient evidence of the superiority of our method.

---

### Official Review · Reviewer_giMg · 2023-07-08

**Soundness:** 3 good
**Presentation:** 3 good
**Contribution:** 3 good
**Rating:** 5
**Confidence:** 2

**Summary:**

This paper addresses the challenge of learning safety constraints for agents in various tasks from expert demonstrations, instead of manually specifying them. The authors extend inverse reinforcement learning (IRL) techniques to the space of constraints, aiming to learn constraints that prevent highly rewarding behavior that the expert could have performed but avoided. However, the constraint learning problem is ill-posed and tends to result in overly conservative constraints. To mitigate this issue, the authors utilize diverse demonstrations from multi-task settings to learn a tighter set of constraints. The method is validated through simulation experiments on high-dimensional continuous control tasks.

**Strengths:**

- The authors develop a multi-task variant of inverse constraint learning, which consists of multiple policy players aiming to maximize task-specific rewards and a constraint player determining a single constraint that all policy players must follow.
- The proposed approach reduces the chances of choosing degenerate solutions.
- The inverse constraint learning problem is formulated as a zero-sum game involving a policy player, who seeks to maximize rewards while adhering to potential constraints, and a constraint player, who selects constraints that impose maximum penalties on the learner compared to the expert.
- The proposed method demonstrates its effectiveness in various continuous control tasks. When applying restricted function classes, the technique can retrieve ground-truth constraints for certain tasks.

**Weaknesses:**

The author argues that previous ICL problems have not been well-formulated, leading to solutions that only allow expert actions. However, it is not clear how the proposed method addresses this issue, aside from utilizing a greater number of demonstrations from a multi-task setting.
It would be interesting to determine whether the proposed method can work for tasks beyond maze locomotion, which lack complex interactions with the environment. For example, in manipulation tasks, safety constraints are more critical as the robot must interact with and alter the environment.

**Questions:**

The author could better address the concerns mentioned in the weaknesses section. It is worth noting that the reviewer is not well-versed in this field and does not specialize in IRL or ICL problems. Thus, the overall rating may be subject to change based on feedback from other reviewers and the author's rebuttal.

**Limitations:**

Consider removing Section 4.2.
A more intuitive explanation of Figure 2 would be helpful.

---

> ### Author Rebuttal · Authors · 2023-08-10
>
> We thank the reviewer for their thorough description of our work. Responding to the concerns raised:
>
> W1: Intuitively, if we see a more diverse set of behaviors from the expert by using multi-task data, our constraint learner will spuriously forbid fewer states. Importantly, it is not the greater number of demonstrations that matters for dealing with ill-posedness so much as it is the greater diversity of behavior. For example, building on the reviewer’s manipulation example, if we only see data from a single pick-and-place task, single-task ICL could recover a constraint that prevents the learner from placing an object anywhere else on the table. As long as these sort of “complex interactions with the environment” are encoded in the state of the agent (i.e. we are not in the partially observed setting), we believe our approach would be applicable.
>
> Please also see our global response on experimental complexity.
>
> While we agree that focusing on manipulation specific applications would be interesting, our current experiments are equally high-dimensional (e.g. for our ant experiments, the state-space is 27 dimensional and the action space is 8 dimensional, compared to a 7 DoF manipulator) and we show results where are able to recover workspace constraints. More explicitly, given that we can learn a position constraint for a high-dimensional ant agent, we should also be able to learn constraints on the end effector of a manipulator). Similarly, if we are able to handle data that comes from the agent navigating to different parts of the maze, placing a block in different locations / goals should not be that different. Hence we see no conceptual reason why the algorithm would not scale to manipulation.
>
> Figure 2: We would be more than happy to add more description of this figure. At a high level, $\rho_{\Pi}$ refers to the space of occupancy measures (i.e. the set of state-action distributions of all policies in our policy class). The reason we chose this representation was that a constraint is simply a hyperplane in this space (the red line labeled $\langle \rho_{\pi}, c^* \rangle = 0$). To uniquely determine a line in $\mathbb{R}^d$, we need $d$ points on the line – these are what the green dots / expert policies $\pi_E^1$ and $\pi_E^2$ correspond to. We need two policies as our visitation distributions are two-dimensional (e.g. a two-state MDP). The technical conditions in Lemma 3.2 are the more formal version of the above statements (e.g. relint meaning the points are on the line rather than on the boundary of the space).

---

> > ### Author Response · Authors · 2023-08-17
> >
> > We thank the reviewer for their thoughtful comments. As we get closer to the end of the discussion period, please let us know if we were able to answer all of your questions or if there is anything else we could clarify.

---

> > > ### Comment · Reviewer_giMg · 2023-08-21
> > > **Reply to Authors**
> > >
> > > Thanks for your reply. I have no further questions. As mentioned in the original review, I have never worked on this field so that I am not confident for the comments for your paper. I believe that the final decision will be more dependent on other reviewers.

---

### Author Rebuttal · Authors · 2023-08-10

We thank all reviewers for their carefully considered feedback.

**Limited Experiments:**

We would like to begin by noting that, in comparison to many of the standard benchmarks in safe RL, our considered tasks are higher dimensional (e.g. our ant-based tasks are higher dimensional than every task in the standard Safety Gym benchmark) and we are successfully solving them without knowing the ground-truth constraint. Furthermore, much of the prior work in ICL has performed experiments purely on gridword problems [Vazquez-Chanlatte et al., Scobee and Shastry et al., McPherson et al.] or used simple constraint function classes like axis-aligned rectangles [Chou et al.], while we perform experiments with deep neural networks on continuous control problems.

That being said, we agree with the reviews that it would only make our paper stronger if our multi-task experiments were of same dimensionality as our single-task experiments and therefore implemented our algorithm on the D4RL AntMaze task (rather than the PointMaze task we included in our initial draft). On this significantly harder problem, we were able to recover all of the walls correctly (within a single iteration) and therefore learn policies that match expert performance and safety. We also performed multiple iterations of ICL to show that our learned constraint is stable (and therefore does not require the early stopping that single-task ICL approaches need). We believe this is strong evidence that our method is able to scale to complex, high-dimensional, multi-task problems. Furthermore, we note that our multi-task method is the only method which correctly recovers the ground truth constraint compared to
a variety of baselines (e.g. the max, mean, or individual single-task constraints) which are overly conservative, highlighting how our focus on the multi-task setup has produced a more practical method than the prior, single-task art. See Figures 5/6 in PDF for full results.

**Relationship to the work of Chou et al.:**

As we pointed out in our related work section, our problem setup is quite similar to the setup of Chou et al., in that we assume access to both safe demonstrations and task rewards. However, algorithmically, we believe our techniques are more general. Chou et al. use variants of hit-and-run sampling to generate low-cost, safe trajectories (a non-standard algorithm for non-convex control problems), while we make no assumptions about the particular constrained RL method one uses. This allows us to learn deep feedback control policies, rather than open-loop trajectories. Their “gridded” constraint learning methods require the use of integer / mixed-integer program solvers while their “parametric” constraint learning methods require restrictive constraint classes (e.g. “axis-aligned hyper-rectangles”) or solving mixed integer feasibility programs. In contrast, we only require the ability to solve a classification problem, making it clear how to learn deep constraint networks using our method. This is why we described our approach as being more “likely to scale to realistic problems” in our writeup: we can leverage flexible function approximators and modern deep reinforcement learning algorithms. As Chou et al. wrote in Sec. 8.4 of their paper, “To scale to high-dimensional constraint spaces, we assume a known, relatively simple parameterization, which is in general not the case for real constraints.” We also note that, given the nonstandard constrained policy optimization and constraint learning methods Chou et al. propose using and the fact that they did not release their code, it would be quite an undertaking for us to re-implement their methods on the problems we consider.

We are able to prove policy performance / safety guarantees for policies learned via our recovered constraint, while we were unable to find the analogous guarantees in the work of Chou et al. In essence, the theory of Chou et al. focuses on constraint recovery which, as we discuss in Sec 3.5, requires a very large number of diverse demonstrations, outside of problems with the relatively simple constraint structures they consider. On a more technical level, we do not require the restrictive Lipschitzness assumptions they make in their analysis section (which again reflects the fact that they are focused on the impossibly high goal of perfect constraint recovery). Also, in theory, Chou et al. require “boundedly suboptimal” demonstrations, while we, as a reviewer points out, do not.

In short, we believe that our writeup captures what is essential to the multi-task inverse constraint learning problem, and allows the reader to pick the particular set of solvers that are most effective on the particular instance they are interested in. Furthermore, we are also able to provide strong guarantees on the performance / safety of the policy optimized under the learned constraint, while Chou et al. seem to focus on a goal that is somewhat of a red herring on realistic problems.


**Expert Optimality Requirement:**

An insight we did not fully appreciate until it was brought to our attention by a reviewer was that nowhere in our formalism do we actually require the expert to be the (soft) optimal policy under the ground truth constraint, as all prior works we know of in the ICL domain [McPherson et al., Chou et al.] do. Of course, if the expert data is extremely suboptimal, it is hard for any method to infer a reasonable constraint. However, in contrast to prior work, our theory does not break down even in this case (as we focus on policy performance / safety rather than perfect constraint recovery) and still provides rigorous guarantees.

To empirically validate this point, we redid our single-task experiments using an expert that is far from optimal on the task. Using this data, we are able to learn a constraint that induces a policy that is as safe and significantly more performant than the expert. See Figures 7/8 PDF for full results.

---

### Decision · Program_Chairs · 2023-09-21

**Decision:**

Accept (poster)

**Comment:**

The reviewers generally found the technical contribution strong, but had some concerns about the strength of the experimental evaluation, including the absence of comparison to prior work, absence of real-world experiments, and the limited complexity of the experimental settings. The authors addressed the former concern with additional experiments. The authors responded to the latter concern by pointing to prior work that solved problems of lower complexity and including additional experiments on a more challenging task that fit the studied setting of multi-task demonstrations, which showed their method performing well. The reviewers generally were satisfied by these responses.